# EDUCE: Explaining model Decisions through Unsupervised Concepts Extraction

## Abstract

Providing explanations along with predictions is crucial in some text processing tasks. Therefore, we propose a new self-interpretable model that performs output prediction and simultaneously provides an explanation in terms of the presence of particular concepts in the input. To do so, our model's prediction relies solely on a low-dimensional binary representation of the input, where each feature denotes the presence or absence of concepts. The presence of a concept is decided from an excerpt i.e. a small sequence of consecutive words in the text. Relevant concepts for the prediction task at hand are automatically defined by our model, avoiding the need for concept-level annotations. To ease interpretability, we enforce that for each concept, the corresponding excerpts share similar semantics and are differentiable from each others. We experimentally demonstrate the relevance of our approach on text classification and multi-sentiment analysis tasks.

## 1 Introduction

While deep learning models are powerful tools to perform a large variety of tasks, their predictive process often remains obscure. Understanding their behavior becomes crucial. This is particularly true with text data, where predicting without justifications has limited applicability. There has been a recent focus on trying to make deep models more interpretable, see for example (Ribeiro et al., 2016; Bach et al., 2015; Shrikumar et al., 2017; Simonyan et al., 2014; Sundararajan et al., 2017). Specifically, methods have been recently proposed to provide such explanations *simultaneously* with the prediction. For example, (Lei et al., 2016; Yu et al., 2019; Bastings et al., 2019) select subsets of words in the input text that can account for the model's prediction (called rationales). In this work, we propose a model that provides an explanation based on the absence or presence "concepts" that are automatically discovered in the texts.

Suppose we ask a user to tell what is the category (or class) of the top text of Figure 1 (text $x$). She detects that the words *the government said* relate to a specific concept that is present in $x$, a concept she also detect in the words *made official what* in text $x'$ . Let us call this concept "politics" in the remainder. She also notes that the words *retails sales bounced back* refer to a concept different from the previous one. Let us call this other concept "economy". As the text is concerned with politics and economy she infers that its category is Business. Said otherwise, she detects excerpts that relate to particular concepts and decides on the text category based on the detected concepts.

Similarly, our paradigm assumes that an explanation of a model's prediction is understandable if it relies on a few concepts, where each concept relates to parts of text (referred to as *excerpts*) that are semantically consistent across multiple texts. Our methodology is as follows. First, our model encodes the input text into a binary low-dimensional vector, where each value (0 or 1) is computed from an excerpt and denotes the presence or absence of a specific concept in the input. Then, the prediction is made from this binary vector alone, allowing an easy interpretation of the decision in term of presence/absence of concepts.

While we set a maximal value of the number of concepts (which is the dimensionality of the binary representation), the concepts are **unsupervised and not defined a priori**. The model automatically determines them in a way that eases interpretation: each concept is encouraged to be semantically consistent and not to overlap with other concepts. Therefore, extracted excerpts for a concept must be discriminative of that concept only and share similar semantics. This is enforced through a concept consistency and separability constraint added as an auxiliary loss in the learning optimization problem.

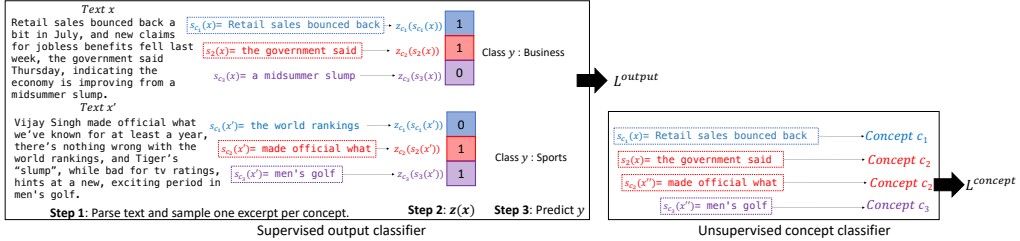

Figure 1: Illustration of EDUCE's prediction process on AGNews topic classification dataset (Zhang & LeCun, 2015) examples. The different steps are explained in the text.

As a result, each discovered concept can be understood from the corresponding excerpts extracted that activate its appearance: in our previous example, the meaning of the first concept the user identifies is inferred from the excerpts she detected for that concept in $x$ and $x'$, i.e. *the government said, made official what*. Looking at these excerpts, we identify that concept as politics.

Our idea relates to Latent Dirichlet Allocation (LDA) (Blei et al., 2003), where text documents are described by a set of topics that are semantically consistent. However, LDA builds a probabilistic model of text generation, whereas our goal is to discover and define latent concepts that are relevant for a prediction task at hand. In comparison to rationale-based text processing models (Lei et al., 2016; Yu et al., 2019; Bastings et al., 2019), we rely on a different paradigm: our model's prediction is based on the absence or presence of discovered concepts, and makes no direct use of the words captured as excerpts. We do so to ease interpretation of the prediction. This makes the interpretation of a different nature than these methods, and simpler to understand for the user.

Our contribution with this work is a new self-interpretable model that predicts solely from the presence/absence of concepts in the input. Concepts are learned unsupervisingly, and described by a semantically consistent set of excerpts. We experiment on three text categorization tasks and a multi-aspect sentiment analysis task, compare our model's performance with state-of-the-art prediction models, and demonstrate its interpretability. Note that an instance of our model for image processing is described in the supplementary Section E with illustrative experiments.

## 2 THE EDUCE MODEL

We present our model, called EDUCE for Explaining model Decisions through Unsupervised Concepts Extraction, using a multi-class classification task, but our method can be used to perform regression or multi-label classification. We consider a training dataset $\mathcal{D}$ of inputs $(x_1, ...., x_N)$, $x_n \in \mathcal{X}$ and corresponding labels $(y_1, ..., y_N), y_n \in \mathcal{Y}$.

Contrarily to back-box models that use complex computations over low-level input features to predict the output class $y$, our objective is to enforce the model to map an $x$ input to an easy-to-interpret representation $\boldsymbol{z}(x) \in \{0, 1\}^C$, on which the output prediction relies. EDUCE's inference process consists in the following steps. **Step 1**: For each concept $c$ (i.e. each dimension of $\boldsymbol{z}(x)$), compute $p_\gamma(s|x, c)$: the probability of each excerpt $s$ in $x$ to be selected. Sample a unique excerpt $s_c(x) \sim p_\gamma(s|x, c)$ per concept (per dimension of $\boldsymbol{z}(x)$). Note that the same excerpt can be selected for multiple concepts. **Step 2**: Decide on each value $z_c(x)$ of the representation $\boldsymbol{z}(x)$ by sampling from $p_\alpha(z_c|s_c(x), c)$. Given $s_c(x)$, $z_c(s_c(x)) = 1$ means concept $c$ is detected as present and $z_c(s_c(x)) = 0$ means it is absent. **Step 3**: Predict the output class $y$ from $\boldsymbol{z}(x) = (z_1(s_1(x)), ..., z_C(s_C(x)))$.

We do not have concept-level annotations. To ensure semantic consistency and prevent overlap of the concepts, we jointly train a concept classifier to recognize, for every excerpt $s_c(x)$ such that $z_c(s_c(x)) = 1$, the concept (i.e. the dimension) it was extracted for: the label for each $s_c(x)$ is simply $c \in [1, C]$. Figure 1 illustrates these steps, which we detail below.

### 2.1 GENERATING PREDICTIONS AND EXPLANATIONS

**Step 1: Extract a unique excerpt** $s_c(x)$ **per concept** $c$. Given an input sentence $x : w_1, ..., w_M$, where each $w_i$ is a word, a excerpt $s$ consists of a span of consecutive words of flexible size between 3 and 10 words: $s = w_k, ..., w_{k+l}, 3 \leq l \leq 10$. For each concept, we extract a unique excerpt,

defined by its first and last word. Our extraction process is very similar to the one proposed in Question-Answering models (e.g Devlin et al. (2019)). An excerpt is sampled by first sampling its *start word* and then its *end word* conditioned on the start word.

We denote the probability over each excerpt $s$ to represent concept $c$ in $x$ as $p_\gamma(s|x, c)$ and we write as the product of (i) $p_{start}(w_k|x, c)$, the probability for $s$'s first word to be the start word of the excerpt to extract and (ii) $p_{stop}(w_{k+l}|x, c, w_k)$ the probability of $w_{k+l}$ to be the stop word given the start word $w_k$: $p_\gamma(s|x, c) = p_\gamma(w_k, ..., w_{k+l}|x, c) = p_{stop}(w_{k+l}|x, c, w_k)p_{start}(w_k|x, c)$. We parametrize $p_{start}(w_k|x, c)$ and $p_{stop}(w_{k+l}|x, c, w_k)$ using recurrent neural networks. First we feed the entire input sentence $x : w_1, ..., w_M$ through a bidirectional LSTM, and we represent each word $w_i$ in the sentence as the concatenation of the forward pass and backward pass hidden states for that word: $h_k = [\overrightarrow{h_k}, \overleftarrow{h_k}], \forall k \in [1, M]$. Second, each $h_k$ is fed to a linear layer with parameters $\gamma_{start}$ which outputs a score for $w_k$, followed by the softmax activation function over all possible words to have the probability distribution over each word to be the *start word*:

$$p_{start}(w_k|x, c) = \frac{\exp(\gamma_{start} \cdot h_k)}{\sum_{k=1}^M \exp(\gamma_{start} \cdot h_k)}, \ k = 1, ..., M. \tag{1}$$

Using this distribution, we can sample a specific start word $w_{start} \sim p_{start}(w_k|x, c)$. Third, we feed again the vectors $h_k$ to another linear layer with parameters $\gamma_{stop}$ that gives a score for each word to be the *stop word*. We mask out these scores before taking the softmax, such that the probabilities for $w_{start+l}$ with $l < 3$ and $l > 10$ to be the *stop word* are 0.

$$p_{stop}(w_{start+l}|x, c, w_{start}) = \begin{cases} \frac{\exp(\gamma_{stop} \cdot h_{start+l})}{\sum_{l=3}^{10} \exp(\gamma_{stop} \cdot h_{start+l})}, \ l \geq 3 \text{ or } l \leq 10 \\ 0, \ l < 3 \text{ or } l > 10. \end{cases}$$

We then sample a specific stop word $w_{stop} \sim p_{stop}(w_{start+l}|x, c, w_{start})$[1]. We now have the start word $w_{start}$ and stop word $w_{stop}$. The corresponding excerpt is $w_{start}, ..., w_{stop}$ and is represented as a fixed size vector $s_c(x)$ by computing the average of (pre-trained, fixed) words embeddings vectors of $w_{start}, ..., w_{stop}$. Hence sampled excerpts $s_c(x)$ belong to $\mathbb{R}^d$, where $d$ is the dimension of embedding vectors.

This process is illustrated in Figure 1: For text $x$, the excerpt *Retails sales bounced back* is selected for concept $c_1$, *the government said* for concept $c_2$ and *a midsummer slump* for concept $c_3$.

**Step 2: For each $c$, from the excerpt $s_c(x)$, decide on the value $z_c(x)$ denoting the presence/absence of concept $c$.** Each extracted excerpt (one per concept) is processed to decide on the absence or presence of each concept in $x$. Specifically, for each concept $c$ we take the dot product of a weight vector $\alpha_c \in \mathbb{R}^d$ with $s_c(x)$, followed by a sigmoid activation function, in order to obtain the Bernoulli probability

$$p_\alpha(z_c = 1|s_c(x), c) = \sigma(\alpha_c \cdot s_c(x)) \tag{2}$$

This is the probability that $s_c(x)$ extracted from $x$ for concept $c$ triggers the presence of concept $c$. The binary vector $\mathbf{z}(x)$ is obtained by independently sampling each dimension: $\forall c \ z_c \sim p_\alpha(z_c|s_c(x), c)$.

This step can be seen as $C$ independent binary classifiers, each of them deciding on the presence of a particular concept. We illustrate this in Step 2 in Figure 1: For text $x$, the excerpt *Retails sales bounced back* activates the presence of concept $c_1$, *the government said* activates the presence of $c_2$ but *a midsummer slump* does not activate the presence of $c_3$.

**Step 3: Predict $y$ from $\mathbf{z}(x)$.** As shown in Step 3 of Figure 1, given an input $x$, the prediction of the output $y$ is made solely from the intermediate binary representation $\mathbf{z}(x)$. We use a linear classifier without bias, parametrized by a weight matrix $\delta \in \mathbb{R}^{|\mathcal{Y}| \times C}$ followed by a softmax activation function, returning $p_\delta(y|\mathbf{z}(x))\forall y \in \mathcal{Y}$.

**Output classification training objective.** The parameters $\{\gamma, \alpha, \delta\}$ are learned in a end-to-end manner by minimizing cross-entropy, which writes for each $x$ as:

$$\mathcal{L}^{output}(x, y, \delta, \alpha, \gamma) = \mathbb{E}_{\mathbf{s}(x) \sim p_\gamma}[\mathbb{E}_{\mathbf{z}(x) \sim p_\alpha}[\mathcal{L}^{output}(y, \mathbf{z}(x), \delta)]]$$
$$= \mathbb{E}_{\forall c \ s_c(x) \sim p_\gamma(s|x, c)}[\mathbb{E}_{\forall c \ z_c \sim p_\alpha(z_c|s_c(x), c)}[-\log p_\delta(y|\mathbf{z}(x))]]. \tag{3}$$

We give details about the optimization in Section 2.3.

---

[1] We also prevent stop words to be outside of the text length.

## 2.2 UNSUPERVISED DISCOVERY OF LATENT CONCEPTS

To ease interpretation of the concepts' meaning, we want each dimension $z$ to correspond to semantically consistent concepts. In other words, for a given concept $c$ (i.e. a given dimension $z_c$), for all inputs where that concept is present (i.e. $\forall x \; s.t. \; z_c(s_c(x)) = 1$), the corresponding excerpts $s_c(x)$ should share common semantics. While the $C$ binary classifiers deciding on the presence of each concept $c$ (Step 2) enforce consistency within the set of extracted excerpts $s_c$, there is no guarantee that the concepts are not overlapping (i.e. are separable). We want concepts to be separable so that each concept captures a particular notion. Excerpts extracted for concept $c$ must be distinguishable from the excerpts extracted for another concept $c'$. If we take the example of Figure 1: In Step 2 the excerpt *Retails sales bounced back* is identifiable as relating to economy/finance, contrarily to *the government said, made official what* which trigger the concept politics.

We rephrase our desiderata for consistency and separability as classification of the excerpts, with as many classes as the number of concepts. An external classifier should recognize that elements in the set $\{s_c(x)|\forall x \; z_c(s_c(x)) = 1\}$ belong to concept $c$, and $\{s_{c'}(x)|\forall x \; z_{c'}(s_{c'}(x)) = 1\}$ belong to concept $c'$ and not $c$. Recall that **we do not have labels for the concepts**. The ground-truth label of each extracted excerpt $s_c(x)$ is the **index** $c$ of the dimension of $z(x)$ for which it was extracted. We learn a linear *concept classifier* without bias, parametrized by a weight matrix $\theta \in \mathbb{R}^{|C| \times d}$ followed by a softmax activation function, returning $p_\theta(c'|s_c(x)), \forall c' \in [1, C]$.

The concept classifier is trained by minimizing cross-entropy, where the label of each excerpt $s_c(x)$ the index of the concept for which it was extracted (i.e. $c$):

$$\mathcal{L}^{concept}(x, \theta, \alpha, \gamma) = \mathbb{E}_{s(x) \sim p_\gamma}[\mathbb{E}_{z(x) \sim p_\alpha}[\mathcal{L}^{concept}(z(x), s(x), \theta)]] \qquad (4)$$

$$= \mathbb{E}_{s(x) \sim p_\gamma}[\mathbb{E}_{z(x) \sim p_\alpha}[\sum_c -z_c(s_c(x)) \log p_\theta(c|s_c(x))]],$$

where $s(x)$ refers to the set of excerpts extracted for each concept: $s(x) = \{s_c(x), c = 1, ..., C\}$. The loss considers concepts that are present in $x$ (i.e. $z_c(s_c(x)) = 1$). The role of the concept classifier is to further enforce concepts consistency and to prevent overlap of concepts, and is jointly train with the rest of the model.

Note that adding a sparsity constraint on the number of concepts present in the inputs enforces semantic consistency, as sparse coding has been shown to induce useful, interpretable representations (Bengio et al., 2013a; Mairal et al., 2010). We experimentally demonstrate that a sparsity constraint is not sufficient, and can harm output prediction performance. On the opposite, with our concept classifier, sparsity is encouraged as $\mathcal{L}^{concept}$ depends on the concepts that are present, but if concepts are consistent and separable $\mathcal{L}^{concept}$ can be low without harming task performance.

## 2.3 EDUCE OBJECTIVE FUNCTION AND OPTIMIZATION

We jointly learn the *concept classifier* and *output classifier* and our objective function is the sum Equations 3 and 4:

$$\mathcal{L}(x) = \mathbb{E}_{s(x) \sim p_\gamma}[\mathbb{E}_{z(x) \sim p_\alpha}[\mathcal{L}^{output}(y, z(x), \delta) + \lambda \mathcal{L}^{concept}(z(x), s(x), \theta)]] \qquad (5)$$

where $\lambda$ controls the strength of the concept consistency constraint wrt the output prediction. The loss $\mathcal{L}(x)$ is differentiable, therefore the gradients with respect to the output classifier's weights ($\delta$) and concept classifier's weights ($\theta$) can be computed and back-propagated. However, the gradients with respect to the parameters of the excerpts extraction ($\gamma$) and the Bernoulli distribution over presence of concepts ($\alpha$) pose an issue due to the sampling of these discrete random variables in Steps 1 and 2. As the explicit computation of the expectation involves expensive summations over all possible values of $s(x)$ and $z(x)$, we resort to Monte-Carlo approximations of the gradient (Sutton & Barto, 1998), with the loss $\mathcal{L}(x)$ used as reinforcement signal. The gradient derivation is provided in Supplementary Material. Our code will be released upon acceptance.

Note that the concept classifier encourages concept consistency and concept separability. The discovered concepts are not guaranteed to align with human defined concepts. This is a general challenge in learning explainable/interpretable models in an unsupervised setting. We believe it is a reasonable assumption to employ a linear concept classifier over pre-trained word embeddings such that it linear separates the word embedding space.

## 3 RELATED WORK

Our work relates to topic models, especially to Latent Dirichlet Allocation (LDA) (Blei et al., 2003) where each document is modeled as a mixture of topics and each word relates to one or more topics. While LDA has been recently combined with recurrent neural networks (Zaheer et al., 2017), the methodology remains different. LDA learns the parameters of a probabilistic graphical model of text generation while our goal is to build an interpretable prediction model that relies on binary presence/absence of concepts.

Some existing works interpret an already trained model, typically using perturbation and gradient-based approach, (Ribeiro et al., 2016; Bach et al., 2015; Shrikumar et al., 2017; Simonyan et al., 2014; Sundararajan et al., 2017) Alvarez-Melis & Jaakkola (2017) design a model that detects input-output pairs that are causally related. Contrarily to these methods, our work falls in the domain of self-interpretable models, which produce an explanation simultaneously with their prediction. Lei et al. (2016); Yu et al. (2019); Bastings et al. (2019) develop interpretable models for NLP tasks by selecting rationales, i.e. parts of text, on which a consequent model bases its prediction. A rationale acts as a justification supporting the prediction and is different from our definition of excerpt. In EDUCE, excerpts support the attribution of presence of concepts, and this attribution supports the prediction. Goyal et al. (2019) propose visual explanations of a classifier's decision and Alaniz & Akata (2019) use an observer-classifier pair model, whose prediction can be exposed as a binary tree. Contrarily to ours, their model does not provide a local explanation based on parts of the input. Quint et al. (2018) extend a classic variational auto-encoder architecture with a differentiable decision tree classifier. However, their methodology is different and they only experiment on image data.

Most related to our work are methods providing concept-based explanations. Alvarez Melis & Jaakkola (2018) learn a self-explainable classifier that takes as input a set of concepts extracted from the original input. They define a set of desiderata for what is an interpretable concept, but simply represent extracted concepts as an encoding of the input learned with an auto-encoding loss. Kim et al. (2018) learn concept activation vectors, and Zhou et al. (2018) build on Bau et al. (2017) to propose a method that generates visual explanations of a classifier, in the form of semantically interpretable components with a corresponding label and heatmap. However, in these models, the final classifier is already trained and fixed while EDUCE is learned end-to-end. More importantly, these works need concepts to be predefined from human annotations, while EDUCE is unsupervised. Ghorbani et al. (2019) propose a method based on unsupervised concept discovery to explain a model a posteriori. We differ from their work as EDUCE is trained end-to-end to classify solely based on the discovered concepts, which presence is discriminant, resulting in a fully interpretable model. Finally, we focus on text data while the aforementioned works experiment on image data.

## 4 TEXT CLASSIFICATION EXPERIMENTS

We experiment on three text classification datasets. The **DBpedia** ontology classification dataset (Zhang et al., 2015) was constructed by picking 14 non-overlapping categories from DBpedia 2014 (Lehmann et al., 2015). There are 14 classes. We use $56,000$ examples of the train dataset for training, and $56,000$ for validation. For testing, we use $7,000$ examples of the test dataset (using stratified sampling). Also in Zhang et al. (2015), the **AGNews** topic classification dataset was constructed from the AG dataset's 4 largest categories. There are 4 classes. We separate the training set into $84,000$ training samples and $24,000$ validation samples and report results on the full test dataset. We also experiment on the Stanford Sentiment Treebank (**SST**) (Socher et al., 2013) that includes 5 sentiment classes. For DBPedia and AGNews we use fixed word vectors trained on Common Crawl (Grave et al., 2018). For SST, we use fixed Glove word vectors (Pennington et al., 2014).

**Comparative models.** Since we are concerned with building self-interpretable models, we do not compare with the methods presented in Section 3 that explain a model a posteriori. Rather, for comparison, we report the performance of (i) using a Bidirectional LSTM on the full text and predict directly from its hidden state (Baseline) (ii) the No Concept Loss model that uses a binary discrete intermediate representation but no concept classifier, it corresponds to $\lambda = 0$ and its training objective is simply $\mathcal{L}^{output}$ and (iii) adding a $L_1$-norm sparsity constraint to the No Concept Loss model (referred to as No Concept Loss + $L_1$), i.e. Equation 5 changes to:

$$\mathcal{L}(x) = \mathbb{E}_{s(x) \sim p_\gamma}[\mathbb{E}_{z(x) \sim p_\alpha}[\mathcal{L}^{output}(y, z(x), \delta) + \lambda_{L_1}|z|]]. \tag{6}$$

Therefore both No Concept Loss and No Concept Loss + $L_1$ models extract excerpts and use a

| Data | Model | Output Acc. (%) | A Posteriori Concept Acc. (%) |
|---|---|---|---|
| DBPedia | EDUCE | $97.0 \pm 0.1$ | $\mathbf{82.4 \pm 0.8}$ |
| | No Concept Loss | $97.4 \pm 0.1$ | $25.9 \pm 0.6$ |
| | No Concept Loss+$L_1$ | $96.5 \pm 0.2$ | $44 \pm 2.6$ |
| | Baseline | $\mathbf{98.75 \pm 0.0}$ | n/a |
| AGNews | EDUCE | $87.5 \pm 0.2$ | $\mathbf{78 \pm 6.5}$ |
| | No Concept Loss | $88.2 \pm 0.1$ | $31.0 \pm 0.7$ |
| | No Concept Loss+$L_1$ | $86.3 \pm 0.7$ | $56 \pm 3.2$ |
| | Baseline | $\mathbf{92.08 \pm 0.1}$ | n/a |

Table 1: Test performance on DBPedia and AGNews (mean $\pm$ SEM).

binary intermediate representation, but are trained without concept loss.

**Hyperparameters selection.** We tried different values of hyperparameters (details in supplementary Section B.1) and for each set of hyperparameters, we run 5 different random seeds. We choose the best hyperparameters set using its average validation performance over the 5 random seeds.

**Metrics.** We report the test output accuracy (Output Acc.) over the task. Evaluating the relevance of discovered concepts is a challenge in the unsupervised setting. To measure how well the discovered concepts align with human defined concepts one would need human experiments, which we consider as future work. However, we evaluate how consistent and separable the discovered concepts are, as a proxy to measure how easy it would be for a user to interpret them. For EDUCE, we can compute the accuracy of the concept classifier on the test data. However, this should be low for models No Concept Loss and No Concept Loss + $L_1$ that do not train with a concept loss. Therefore, to evaluate interpretability, we report an *a posteriori* concept accuracy: after training, for each model (including EDUCE), we gather the excerpts extracted ($z_c = 1$) in the test data. We separate these excerpts into two sets (training and testing, note that these are both generated from the test data). We train a new, separate linear concept classifier *a posteriori* on the extracted excerpts to evaluate and we report its performance as a posteriori concept accuracy (A Posteriori Concept Acc.).

**Quantitative results on DBPedia and AGNews.** Table 1 reports test performance on the DBPedia and AGNews datasets for $C = 10$ concepts. We report results with values $\lambda = 0.1$ and $\lambda_{L_1} = 0.1$ that give best trade-off performance between output and concept accuracies. Complete results are available in supplementary Table 4. For all metrics we report the mean and standard error of the mean (SEM) over the training random seeds. Table 1 shows that the Baseline model outperforms the other models that all employ a binary discrete encoding of the input. This is expected, and shows the trade-off between interpretability and output accuracy. In terms of output accuracy, EDUCE is comparable with its counterpart that does not encourage concept consistency (No Concept Loss), loosing only $0.4\%$ on DBPedia and $0.7\%$ on AGNews, and as expected outperforms it in terms of concept accuracy. Adding an $L_1$ constraint to the No Concept Loss model increases the consistency of the concepts (as per A posteriori Concept Accuracy values) yet is largely outperformed by EDUCE on that metric, and output accuracy is also lower than EDUCE's.

**Interpreting EDUCE on AGNews.** We turn to show how EDUCE's output prediction is easily interpretable. The following results were generated with $\lambda = 0.1$ and $C = 10$ concepts. Figure 2a shows examples of the AGNews test set that were correctly classified by EDUCE. The underlined words correspond to the excerpts extracted for different concepts. Separately, in Figure 2b we show, for each concept detected in the examples of Figure 2a, examplar excerpts extracted from others test documents (excerpts are separated by "/"). We interpret the concepts' meaning as follows: concept 0 maps to informatics notions, concept 2 to corporations/petrol, concept 4 to the notions of investments and finance, concept 6 to sports events and concept 8 to governmental/state affaires. Concept 1 is less clearly defined. Note that in Figure 2b excerpts are consistent yet are extracted in texts from multiple output classes. The excerpts extracted in the examples of Figure 2a are consistent with these interpretations. These results show how easily the classification of any text can be explained by the detection of multiple, relevant, and intelligible concepts. More qualitative examples are in supplementary Section C. Note that supplementary Table 10 shows the concepts extracted by the No Concept Loss model, we see that they form less semantically consistent units and are hard to interpret. Supplementary table 8 shows test examples where the final classifier's prediction was incorrect. In some of these, the predicted class could be considered as an appropriate label, and importantly the concepts that are identified as present in the text are relevant with the set of discovered concepts.

Class Business: *sports retailer jjb yesterday reported a near 25 drop in profits and continuing poor sales , and ended shareholders #39 hopes of a takeover by announcing that a potential bidder had walked away .*

Class World: *bangkok , thailand sept . 30 , 2004 - millions of volunteers led by emergency teams fanned out across thailand on thursday in a new drive to fight bird flu after the prime minister gave officials 30 days to eradicate the epidemic .*

Class Sports: *the spanish government responded to diplomatic pressure from britain yesterday by starting a search for fans who racially abused england players during a quot friendly quot football match with spain .*

Class Sci/Tech: *los angeles ( reuters ) - a group of technology companies including texas instruments inc . &lt txn . n&gt , stmicroelectronics &lt stm . pa&gt and broadcom corp . &lt brcm . o&gt , on thursday said they will propose a new wireless networking standard up to 10 times the speed of the current generation .*

(a) AGNews test examples correctly classified by EDUCE. Underlined set of words are excerpts extracted, one color per concept.

| Concept 0 | software services giant / moonwalk to home / video display chip / downloading music . |
|---|---|
| Concept 1 | upcoming my prerogative video / his ever-growing swimming / launch of a video display chip / illegality of downloading |
| Concept 2 | oil market . / oil giant sibneft / oil prices and / corp . &lt brcm |
| Concept 4 | cash settlement of up to #36 50 million / six-year deal worth about $40 million / the dollar dipped to a four-week low against the euro / five shares , |
| Concept 6 | olympic 100-meter freestyle / sox ' family / olympics should help / athletes were already |
| Concept 8 | frail pope john paul / indian army major shot / unions representing workers / goverment representatives . |

(b) Examples of excerpts that are extracted **accross the test set**, corresponding to the concepts detected in Figure 2a. Colors match the colors used in Figure 2a.

Figure 2: Interpretation of EDUCE prediction through concept analysis.

**SST dataset** Figure 3d shows output accuracy versus a posteriori concept accuracy on the SST dataset, for different different number of concepts $C$ and different values of $\lambda$: Each marker is a different value of $\lambda$, and values are (from left to right markers) $\{0, 0.01, 0.1, 0.5\}$. We do the same for the No Concept Loss + $L_1$ model. Note that the left most markers in all curves correspond to $\lambda = 0$ and $\lambda_{L_1} = 0$, therefore to the model referred to as No Concept Loss. While our goal and model is very different than Lei et al. (2016); Bastings et al. (2019), we report their test scores (obtained using 40% of the text on the SST dataset, we report directly from Bastings et al. (2019)). The a posteriori concept accuracy for Baseline, Lei et al. (2016) and Bastings et al. (2019) is not applicable. Figure 3d confirms our results: EDUCE (in blue, purple and red) overperforms the model No Concept Loss + $L_1$ (in green) on both output and a posteriori concept accuracies, and achieve output accuracy comparable to Lei et al. (2016) (orange line) while having 60% a posteriori concept accuracy. Regarding the effect of the number of concept $C$, for a fixed $C$, choosing the adequate value of $\lambda$ allows the user to balance between final accuracy and concepts' accuracy. Similar performances can be achieved by different number of concepts: $C = 20, \lambda = 0.1$ (dotted purple line, squared marker) achieves $\sim 41.5\%$ final accuracy and $\sim 70\%$ concept accuracy, while $C = 5, \lambda = 0.01$ (plain blue line, diamond marker) achieves $\sim 42.0\%$ final accuracy and $\sim 60\%$ concept accuracy. The smaller the number of concepts, the smaller the value of $\lambda$ which is expected as with less concepts the task of the concept classifier is easier.

Figure 3a shows the empirical frequency of presence of each concept, per output class in the SST dataset, using $\lambda = 0.1$ and $C = 10$ concepts. We see that concepts 3, 5 and 7 are often triggered when a sample is positive or very positive, and on the opposite concept 8 is present in negative/very negative texts. The values of weight matrix $\delta$ of the output classifier in Figure 3b proves that the presence of these concepts is responsible for the output classifier's prediction. We can further analyze the concepts: Each word in SST sentences is annotated with a label for the sentiment it expresses, referred to as *word label*. We do not use them during training or validation, but at test-time this allows us to analyze the repartition of words labels in the set of excerpts extracted for each concept. Figure 3c shows that indeed the excerpts selected for concepts 3, 5 and 7 are mostly composed of positive or very positive words, and the excerpts extracted when concept 8 is triggered are mostly composed of negative or very negative words[2].

---

[2]We do not show the amount of neutral words selected as it squeezes the histograms.

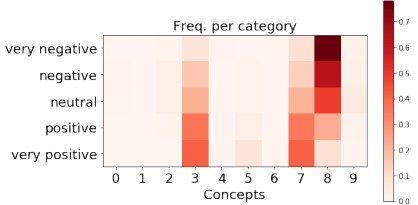

(a) Per class concept frequency.

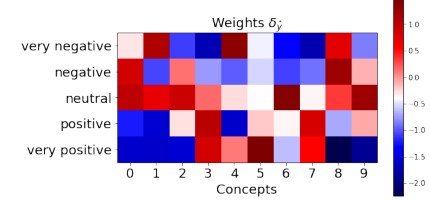

(b) Output classifier weight matrix $\delta$.

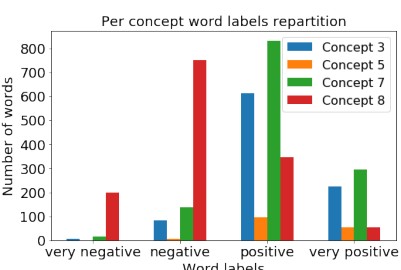

(c) Repartition of words labels for concepts 3, 5, 7, 8.

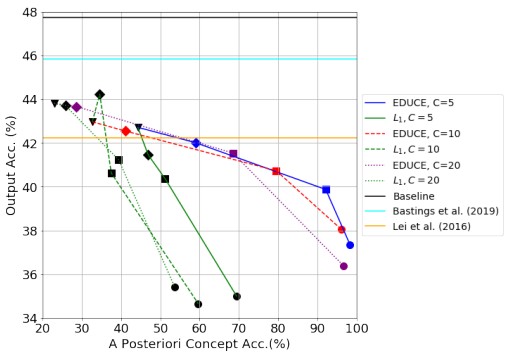

(d) SST test performance, output accuracy vs a posteriori concept accuracy. A posteriori concept accuracy is not applicable for Baseline, Lei et al. (2016) and Bastings et al. (2019).

Figure 3: Results on the SST dataset.

## 5 MULTI ASPECT SENTIMENT ANALYSIS EXPERIMENT

**BeerAdvocate dataset** While our paradigm is different than Lei et al. (2016); Bastings et al. (2019), we also perform a multi-aspect sentiment regression experiment on the pre-processed subset of the BeerAdvocate[3] dataset (McAuley et al., 2012). It consists of $260,000$ beer reviews where ratings for four aspects (look, smell, palate and taste) are given, as well as an overall rating. The reviews are separated into training/validation sets specific to each aspect, and ratings are mapped to scalars in $[0, 1]$. This is a regression task, so we modify the Baseline model and EDUCE in the last layer of the prediction with a sigmoid activation function, and use Mean Squared Error (MSE) instead of cross-entropy for $\mathcal{L}^{output}(y, \boldsymbol{z}(x), \delta)$. McAuley et al. (2012) provided a test set (994 reviews) with sentence-level rationale annotations: different parts of each review are annotated with one (or multiple) aspect label, indicating what aspect it covers (referred to as gold rationales). Importantly, we do not use these gold rationales during training or validation, but use to evaluate our model. Hyperparameters selection is similar to our classification experiments.

We train EDUCE on the prediction of the 5-values vector at once (4 aspects and the overall score, there are therefore 5 scalar ratings to predict) using the $260,000$ reviews. We compare EDUCE's performance with the Baseline that accesses the full text. The test MSE is $0.0089 \pm 0.0$ (mean and SEM accross training seeds) for the Baseline model and $0.0119 \pm 0.0$ for EDUCE using $\lambda = 0.01$ and $C = 10$ concepts, with a corresponding concept accuracy of $92.8 \pm 0.7$. Of specific interest to us is the per-concept precision of gold rationales and the total percentage of selected words. Indeed, Table 3 shows that when predicting the 4 aspects and the overall score, some concepts are capturing a specific aspect: for example, the precision of Concept 7 on the Appearance aspect is $97.09\%$. This is confirmed by Table 2 that shows that excerpts extracted for Concept 7 are mostly related to the color and head (the frothy foam on top of beer) of the beer[4]. Supplementary Table 5 reports examplar excerpts for all concepts.

As in Lei et al. (2016); Bastings et al. (2019) we also train EDUCE on the prediction of each of the first three aspects separately. Supplementary Table 6 reports the precision of gold rationales for each aspect. While the results are less convincing than the models specifically designed for rationales extraction, extracted excerpts match gold rationales to a certain extent.

---

[3]https://www.beeradvocate.com/
[4]Tables 3 and 2 reports results of the best seed for the cross-validated hyperparameters.

| Concept | Appearance | Smell | Palate | Taste |
|---|---|---|---|---|
| Concept 0 | 0.85 | **55.75** | 1.79 | 40.60 |
| Concept 1 | 0.54 | 5.48 | 11.70 | 30.58 |
| Concept 2 | 1.22 | 0.90 | **78.18** | 15.03 |
| Concept 3 | 1.46 | **92.39** | 0.30 | 5.34 |
| Concept 4 | 45.33 | 0.23 | 0.26 | 0.16 |
| Concept 5 | 7.27 | 22.50 | 2.27 | 28.86 |
| Concept 6 | 0.36 | 5.86 | 6.40 | 28.73 |
| Concept 7 | **97.09** | 0.98 | 0.45 | 0.53 |
| Concept 8 | 0.18 | 0.09 | 18.07 | 5.93 |
| Concept 9 | 0.00 | 0.00 | **68.42** | 10.53 |

| | |
|---|---|
| Concept 0 | fruity esters , and/ fruits , caramelized pecans , and/ toffee and caramel accents ,/ coffee and chocolate flavors/ earthy hop resin . |
| Concept 2 | creamy and a/ chewy and rich and drinkability/ creamy mouthfeel that/ smooth and just velvety on the/ thick , and |
| Concept 3 | rich malt scents/ aroma is quite hoppy with big citrus/ tons of different sweet malts , toffe/ smells extremely roasty/ boom of grapefruity |
| Concept 7 | good head and lacing/ beautiful golden-amber color/ creamy tan head/ nice deep brown color/ proud head has settled . nothing |
| Concept 9 | carbonation is graceful/ drinkability is excellent/ mouthfeel is wonderful/ mouthfeel is exemplary/ drinkability : excellent |

Table 3: Per-concept precision of gold rationales (in % for each aspect), trained to predict all 4 aspects and the overall score. In bold we emphasize the concepts which precision is above 50% for an aspect's gold rationales.

Table 2: Examples of excerpts extracted on the Beer test set.

## 6 DISCUSSION AND PERSPECTIVES

We propose a new self-interpretable model, EDUCE, that maps inputs to an easy-to-interpret representation on which the output prediction relies. We experimentally demonstrate its ability (i) to extract semantically meaningful concepts described by consistent excerpts, and (ii) to ground its output prediction on the presence/absence of these concepts in each input. EDUCE generates relevant explanations while retaining high predictive performance. As future direction for research, we contemplate using hierarchical representations, as human notions can often be represented with hierarchical structures, for example as in WordNet (Miller, 1995). Furthemore, our principles can be applied on different types of inputs like images, and we provide initial experiment in Supplementary Section E.

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

## A    DETAILS ON THE LEARNING ALGORITHM

As mentioned in our main paper, when computing Equation 5 the cross-entropy losses of the output classifier and concept classifier are differentiable, therefore the gradients with respect to the weights $\delta$ and $\theta$ can be computed. However, the gradients with respect to the parameters of the LSTM ($\gamma$) and the Bernouilli distribution over presence of concepts ($\alpha$) pose a difficulty because we sample $\forall c \; s_c(x) \sim p_\gamma(s|x,c)$ and $\forall c \; z_c \sim p_\alpha(z_c|s_c(x), c)$.

We use straight-through estimator (Bengio et al., 2013b) in the backward pass for the parameters $\alpha$, and resort to Monte-Carlo approximations of the gradient employed in Reinforcement Learning (Sutton & Barto, 1998) for the parameters $\gamma$. The loss $\mathcal{L}(x)$ serves as reinforcement signal. We weight the two terms of the gradient using a parameter $r$ as follows:

$$\nabla_{\delta,\theta,\gamma,\alpha}\mathcal{L}(x) \widehat{=} (1-r)\nabla_{\delta,\theta,\alpha}\mathcal{L}(x) + r\mathbb{E}_{s(x)\sim p_\gamma}[(\mathcal{L}(x) - b)\nabla_\gamma \log p_\gamma(s(x)|x)] \qquad (7)$$

where $r$ controls the strength of the Reinforcement Learning term and $b$ is the average of the loss, used as control variate.

## B    EXPERIMENTAL DETAILS AND DETAILED RESULTS.

We monitor validation performance and use early stopping. For the comparative models we monitor the output prediction accuracy. For EDUCE we monitor the sum of the output prediction accuracy and the concept classifier accuracy (weighted by the parameter $\lambda$).

## B.1 HYPERPARAMETERS CONSIDERED

In all experiment, we increase the value of $\lambda$ by $10\%$ every epoch. In order to choose the best set of hyperparameters, we take the average value on the 5 random seeds.

**DBPedia and AGNews** We try the following ranges of hyperparameters $\lambda \in \{0.0, 0.001, 0.01, 0.1\}$, $\lambda_{L1} = \{0.0, 0.001, 0.01, 0.1, 1\}$, $r \in \{0.01, 0.1\}$, learning rate $lr = \{0.0005, 0.001\}$. The hidden state of the LSTM is of size 200. We use Adam optimizer (Kingma & Ba, 2014), batches of size 64. For the Baseline model, we try adding weight decay in $\{0.0, 1e-6\}$ and dropout in $\{0.0, 0.1, 0.2\}$.

**SST** We try the following ranges of hyperparameters $\lambda \in \{0.0, 0.001, 0.01, 0.1\}$, $\lambda_{L1} = \{0.0, 0.001, 0.01, 0.1, 1\}$, $r \in \{0.01, 0.1\}$, learning rate $lr = \{0.0002, 0.001\}$ and weight decay in $\{0.0, 1e-6\}$. The hidden state of the LSTM is of size 150. We use Adam optimizer (Kingma & Ba, 2014), batches of size 25. For the Baseline model, we follow Bastings et al. (2019) and use weight decay $1e-6$ and dropout $\{0.5\}$.

**BEER** We try the following ranges of hyperparameters $\lambda \in \{0.001, 0.01, 0.1\}$, $r \in \{0.001, 0.01, 0.1\}$, learning rate $lr = \{0.0005, 0.001\}$. The hidden state of the LSTM is of size 200. We use Adam optimizer (Kingma & Ba, 2014), batches of size 256. We also clip the gradient to the value of 1 and employ an entropy maximization regularization on mean of the Bernoulli distributions $\forall c, p_\alpha(z_c|s_c(x), c)$, weighted by 0.001. We train all models for 250 epochs and use early stopping on the validation set. For the Baseline model on the BEER dataset, we use the same hyperparameters as in Bastings et al. (2019) ($lr = 0.0004$, hidden dimension 200, dropout in $\{0.1, 0.2\}$ and weight decay $1e-6$) and train for 100 epochs.

### B.1.1 DETAILED RESULTS ON AGNEWS AND DBPEDIA

Table 4 details output accuracy, a posteriori concept accuracy, and sparsity (number of concepts presents per input) for different values of $\lambda$ and $\lambda_{L_1}$ considered.

| Dataset | Model | Output Acc. (%) | A Posteriori Concept Acc. (%) | Sparsity |
|---|---|---|---|---|
| | Baseline | $98.75 \pm 0.0$ | n/a | n/a |
| | No Concept Loss | $97.4 \pm 0.1$ | $25.9 \pm 0.6$ | $5.51 \pm 0.1$ |
| | EDUCE $\lambda = 0.001$ | $97.3 \pm 0.1$ | $28 \pm 1.7$ | $5.60 \pm 0.1$ |
| | EDUCE $\lambda = 0.01$ | $97.3 \pm 0.1$ | $39 \pm 3.5$ | $5.39 \pm 0.1$ |
| DBPEDIA | EDUCE $\lambda = 0.1$ | $97.0 \pm 0.1$ | $82.4 \pm 0.8$ | $3.5 \pm 0.2$ |
| | No Concept Loss + $L_1$, $\lambda_{L_1} = 0.001$ | $97.4 \pm 0.1$ | $27 \pm 1.9$ | $5.5 \pm 0.1$ |
| | No Concept Loss + $L_1$, $\lambda_{L_1} = 0.01$ | $97.46 \pm 0.1$ | $28 \pm 2.1$ | $5.12 \pm 0.0$ |
| | No Concept Loss + $L_1$, $\lambda_{L_1} = 0.1$ | $96.5 \pm 0.2$ | $44 \pm 2.6$ | $3.1 \pm 0.1$ |
| | No Concept Loss + $L_1$, $\lambda_{L_1} = 1.00$ | $84 \pm 2.0$ | $77 \pm 2.4$ | $1.31 \pm 0.0$ |
| | Baseline | $92.08 \pm 0.1$ | n/a | n/a |
| | No Concept Loss | $88.2 \pm 0.1$ | $31.0 \pm 0.7$ | $5.2 \pm 0.2$ |
| | EDUCE $\lambda = 0.001$ | $88.4 \pm 0.1$ | $33 \pm 1.6$ | $5.2 \pm 0.2$ |
| AGNews | EDUCE $\lambda = 0.01$ | $88.6 \pm 0.1$ | $40 \pm 2.1$ | $4.6 \pm 0.2$ |
| | EDUCE $\lambda = 0.1$ | $87.5 \pm 0.2$ | $78 \pm 6.5$ | $2.4 \pm 0.2$ |
| | No Concept Loss + $L_1$, $\lambda_{L_1} = 0.001$ | $88.3 \pm 0.2$ | $31.6 \pm 0.7$ | $5.1 \pm 0.2$ |
| | No Concept Loss + $L_1$, $\lambda_{L_1} = 0.01$ | $88.7 \pm 0.2$ | $34 \pm 1.4$ | $4.6 \pm 0.2$ |
| | No Concept Loss + $L_1$, $\lambda_{L_1} = 0.1$ | $86.3 \pm 0.7$ | $56 \pm 3.2$ | $2.2 \pm 0.1$ |
| | No Concept Loss + $L_1$, $\lambda_{L_1} = 1.00$ | $53 \pm 3.2$ | $89 \pm 2.8$ | $0.50 \pm 0.0$ |

Table 4: Test performance on DBPedia and AGNews (mean $\pm$ SEM) for all values of $\lambda$ and $\lambda_{L_1}$.

## B.2 ADDITIONAL RESULTS ON BEER

As in Lei et al. (2016); Bastings et al. (2019), we also train EDUCE on the prediction of each of the first three aspects (the 4th aspect they do not experiment on)[5]. Table 6 reports the precision of gold rationales for that aspect and the total percentage of selected words, values for Lei et al. (2016); Bastings et al. (2019) are taken from Bastings et al. (2019). We use $C = 10$ and $\lambda = 0.1$.

---

[5]Bastings et al. (2019) call the third aspect Taste but they actually refer to Palate.

| Concept 0 | fruity esters , and/ fruits , caramelized pecans , and/ toffee and caramel accents ,/ coffee and chocolate flavors/ earthy hop resin . |
| Concept 1 | delicious and with/ enjoyed this beer/ damn good beer/ nice stout that/ good beer to |
| Concept 2 | creamy and a/ chewy and rich and drinkability/ creamy mouthfeel that/ smooth and just velvety on the/ thick , and |
| Concept 3 | rich malt scents/ aroma is quite hoppy with big citrus/ tons of different sweet malts , toffe/ smells extremely roasty/ boom of grapefruity |
| Concept 4 | octoberfest : say the word and you probably/ 2004 vintage : pours/ 12 ounce bottle acquired in a trade/ i hate to be the guy/ 12oz bottle , best before 9/09 , sampled 5/16/09 |
| Concept 5 | which is fantastic/ it is delicious/ is intensely packed/ and a lovely/ builds a wonderful |
| Concept 6 | 'm really digging/ a damn good/ a super fresh growler/ the better abitas/ a great traditional |
| Concept 7 | good head and lacing/ beautiful golden-amber color/ creamy tan head/ nice deep brown color/ proud head has settled . nothing |
| Concept 8 | easy to drink/ hot summer day/ drinkability easy/ easy to down/ can easily turn this 'strong ale ' into a session |
| Concept 9 | carbonation is graceful/ drinkability is excellent/ mouthfeel is wonderful/ mouthfeel is exemplary/ drinkability : excellent |

Table 5: Examples of excerpts extracted on the Beer test set.

| Model | Appearance | | Smell | | Palate | |
|---|---|---|---|---|---|---|
| | Prec. | Extr. | Prec. | Extr. | Prec. | Extr. |
| EDUCE | $81 \pm 6.9$ | $4.9 \pm 0.5$ | $79 \pm 3.1$ | $4.8 \pm 0.4$ | $50 \pm 6.6$ | $4.1 \pm 0.6$ |
| Lei et al. (2016) | 96.3 | 14 | 95.1 | 7 | 80.2 | 7 |
| Bastings et al. (2019) | 98.1 | 13 | 96.8 | 7 | 89.8 | 7 |

Table 6: Precision (Prec., in %) of gold rationales and percentage of extracted text (Extr. in %) when trained separately on each aspect.

We consider a word as selected if any of the concept selects it, and count it as one selected even if it is selected by multiple concepts. While the results are less convincing than the aforementioned rationales models that are specifically designed for this task, we see that extracted excerpts match gold rationales to a certain extent, especially for the aspects Appearance and Smell. For the Palate aspect, manual inspection shows us that for some seeds, the model defines a concept dedicated to the notion of numbers (e.g. corresponding to excerpts "12 ounce bottle") that is consistent, yet not part of the gold rationale.

## C ADDITIONAL QUALITATIVE EXAMPLES ON DBPEDIA AND AGNEWS

Class Sports: *just imagine what david ortiz could do on a good night ' s rest . ortiz spent the night before last with his baby boy , d ' angelo , who is barely 1 month old . he had planned on attending the red sox ' family day at fenway park yesterday morning , but he had to sleep in . after all , ortiz had a son at home , and he . . .*

Class Sports: *foxborough – looking at his ridiculously developed upper body , with huge biceps and hardly an ounce of fat , it ' s easy to see why ty law , arguably the best cornerback in football , chooses physical play over finesse . that ' s not to imply that he ' s lacking a finesse component , because he can shut down his side of the field much as deion sanders . . .*

Class Sports: *ap - american natalie coughlin won olympic gold in the 100-meter backstroke monday night . coughlin , the only woman ever to swim under 1 minute in the event , finished first in 1 minute , 0 . 37 seconds . kirsty coventry of zimbabwe , who swims at auburn university in alabama , earned the silver in 1 00 . 50 . laure manaudou of france took bronze in 1 00 . 88 .*

Class World: *lourdes , france - a frail pope john paul ii , breathing heavily and gasping at times , celebrated an open-air mass on sunday for several hundred thousand pilgrims , many in wheelchairs , at a shrine to the virgin mary that is associated with miraculous cures . at one point he said help me in polish while struggling through his homily in french . . .*

Class World: *najaf , iraq - explosions and gunfire rattled through the city of najaf as u . s . troops in armored vehicles and tanks rolled back into the streets here sunday , a day after the collapse of talks - and with them a temporary cease-fire - intended to end the fighting in this holy city . . .*

Class World: *kabul , afghanistan - government troops intervened in afghanistan ' s latest outbreak of deadly fighting between warlords , flying from the capital to the far west on u . s . and nato airplanes to retake an air base contested in the violence , officials said sunday . . .*

Class Business: *london ( reuters ) - the dollar dipped to a four-week low against the euro on monday before rising slightly on profit-taking , but steep oil prices and weak u . s . data continued to fan worries about the health of the world ' s largest economy .*

Class Business: *new york ( reuters ) - the dollar extended gains against the euro on monday after a report on flows into u . s . assets showed enough of a rise in foreign investments to offset the current account gap for the month .*

Class Business: *reuters - apparel retailers are hoping theirback-to-school fashions will make the grade amongstyle-conscious teens and young adults this fall , but it couldbe a tough sell , with students and parents keeping a tighterhold on their wallets .*

Table 7: AGNews test examples correctly classified by EDUCE. Underlined set of words are excerpts extracted, one color per concept.

Predicted: Business, True Class: Sci/Tech *hong kong ( reuters ) - dell inc . &lt dell . o&gt , the world ' s largest pc maker , said on monday it has left the low-end consumer pc market in china and cut its overall growth target for the country this year due to stiff competition in the segment .*

Predicted: Business, True Class: World *supporters and rivals warn of possible fraud government says chavez ' s defeat could produce turmoil in world oil market .*

Predicted: Sports, True Class: World *athens , greece - top american sprinters jason lezak and ian crocker missed the cut in the olympic 100-meter freestyle preliminaries tuesday , a stunning blow for a country that had always done well in the event . pieter van den hoogenband of the netherlands and australian ian thorpe advanced to the evening semifinal a day after dueling teenager michael phelps in the 200 freestyle , won by thorpe . . .*

Predicted: Sci/Tech, True Class: Business *san francisco – in the latest of a series of product delays , intel corp . has postponed the launch of a video display chip it had previously planned to introduce by year end , putting off a showdown with texas instruments inc . in the fast-growing market for high-definition television displays .*

Predicted: World, True Class: Sports *athens – the mistakes were so minor . carly patterson ' s foot scraping the lower of the uneven bars . courtney kupets ' tumbling pass that ended here instead of there . mohini bhardwaj ' s slight stumble on the beam .*

Predicted: World, True Class: Sports *thens , aug . 17 - so michael phelps is not going to match the seven gold medals won by mark spitz . and it is too early to tell if he will match aleksandr dityatin , the soviet gymnast who won eight total medals in 1980 . but those were not the . . .*

Table 8: AGNews test examples uncorrectly classified by EDUCE. Underlined set of words are excerpts extracted, one color per concept.

| | |
|---|---|
| Concept 0 | software services giant / moonwalk to home / video display chip / downloading music . |
| Concept 1 | upcoming my prerogative video / his ever-growing swimming / launch of a video display chip / illegality of downloading |
| Concept 2 | oil market . / oil giant sibneft / oil prices and / ipos . public |
| Concept 3 | 100-meter freestyle preliminaries tuesday , a stunning blow for a / the red sox ' family day at / the winner . / entire era . |
| Concept 4 | cash settlement of up to #36 50 million / six-year deal worth about $40 million / the dollar dipped to a four-week low against the euro / five shares , |
| Concept 5 | york , light / boston and texas / london ( reuters / newsday #146 s |
| Concept 6 | olympic 100-meter freestyle / sox ' family / olympics should help / athletes were already |
| Concept 7 | gunfire rattled through the / election . on / democratic coordination or cd as the / wiretapping internet phones to monitor criminals and terrorists is |
| Concept 8 | frail pope john paul / indian army major shot / unions representing workers / government representatives . |
| Concept 9 | lourdes , france / ap - american natalie / new york - / ap - it |

Table 9: Examples of excerpts that are extracted on AGNews for all 10 concepts. Colors match the colors used in Table 7 and Table 8.

| | |
|---|---|
| Concept 0 | lourdes , france / david ortiz could do on a good night ' / ) - monsanto / foaf/loaf and bloom filters have |
| Concept 1 | man who claims gov / install shutters outside their windows / reuters - apparel retailers / downloading music . the ignorance |
| Concept 2 | najaf , iraq - explosions and gunfire / latest veterans committee ballot / steep oil prices / billed its ipo |
| Concept 3 | google inc . ' / the pollution , the daylight , the noise / video display chip / downloading music . the ignorance |
| Concept 4 | lourdes , france / veterans committee ballot / intel corp . has / downloading music . the ignorance |
| Concept 5 | sprinters jason lezak and ian crocker missed / david ortiz could / the season . / wireless networking standard up to 10 times the speed |
| Concept 6 | oil market . / david ortiz could do on a good night ' / profit-taking , but steep oil prices / boston the jury is still out on whether a computer |
| Concept 7 | pilgrims , many in wheelchairs / david ortiz could / factory is never / nobel laureate in medicine |
| Concept 8 | , france - a frail pope / new york ( / world ' s largest economy / maps , figures and endless charts |
| Concept 9 | afp - india ' s tata iron and steel company / quot it hurt / profit-taking , but steep oil / goverment representatives . but i hope you find it |

Table 10: Examples of excerpts that are extracted on AGNews for all 10 concepts without No concept Loss. We see the concepts are inconsistent and hard to parse.

Class Company: *transurban manages and develops urban toll road networks in australia and north america . it is a top 50 company on the australian securities exchange ( asx ) and has been in business since 1996 . in australia transurban has a stake in five of sydney ' s nine motorways and in melbourne it is the full owner of citylink which connects three of the city ' s major freeways . in the usa transurban has ownership interests in the 495 express lanes on a section of the capital beltway around washington dc .*

Class Animal: *the red-necked falcon or red-headed merlin ( falco chicquera ) is a bird of prey in the falcon family . this bird is a widespread resident in india and adjacent regions as well as sub-saharan africa . it is sometimes called turumti locally . the red-necked falcon is a medium-sized long-winged species with a bright rufous crown and nape . it is on average 30–36 cm in length with a wingspan of 85 cm . the sexes are similar except in size males are smaller than females as is usual in falcons .*

Class Plant: *astroloma is an endemic australian genus of around 20 species of flowering plants in the family ericaceae . the majority of the species are endemic in western australia but a few species occur in new south wales victoria tasmania and south australia . species include astroloma baxteri a . cunn . ex dc . astroloma cataphractum a . j . g . wilson ms astroloma ciliatum ( lindl . ) druce astroloma compactum r . br . astroloma conostephioides ( sond . ) f . muell . ex benth .*

Class Album: *stars and hank forever was the second ( and last ) release in the american composers series by the avant garde band the residents . the album was released in 1986 . this particular release featured a side of hank williams songs and a medley of john philip sousa marches . this was also the last studio album to feature snakefinger . kaw-liga samples the rhythm to michael jackson ' s billie jean and did well in europe it is as close as the residents ever got to a bona fide commercial hit .*

Class Written Work: *fire ice is the third book in the numa files series of books co-written by best-selling author clive cussler and paul kemprecos and was published in 2002 . the main character of this series is kurt austin . in this novel a russian businessman with tsarist ambitions masterminds a plot against america which involves triggering a set of earthquakes on the ocean floor creating a number of tsunami to hit the usa coastline . it is up to kurt and his team and some new allies to stop his plans .*

Class Educational Inst.: *avon high school is a secondary school for grades 9-12 located in avon ohio . its enrollment neared 1000 as of the 2008-2009 school year with a 2008 graduating class of 215 . the school colors are purple and gold . the school mascot is an eagle . the avon eagles are part of the west shore conference . they will be moving to the southwestern conference beginning in the 2015-2016 school year .*

Class Artist: *vicky hamilton ( born april 1 1958 ) is an american record executive personal manager promoter and club booker writer ( journalist playwright and screenwriter ) documentary film maker and artist . hamilton is noted for managing the early careers of guns n ' roses poison and faster pussycat for being a management consultant for mötley crüe and stryper a 1980s concert promoter on the sunset strip and a club booker at bar sinister from 2001 to 2010 . [...]*

Class Athlete: *james jim arthur bacon ( birth registered october–december 1896 in newport district — death unknown ) was a welsh rugby union and professional rugby league footballer of the 1910s and ' 20s and coach of the 1920s playing club level rugby union ( ru ) for cross keys rfc and representative level rugby league ( rl ) for great britain and wales and at club level for leeds as a wing or centre i . e . number 2 or 5 or 3 or 4 and coaching club level rugby league ( rl ) for castleford .*

Class Office Holder: *jonathan david morris ( october 8 1804 - may 16 1875 ) was a u . s . representative from ohio son of thomas morris and brother of isaac n . morris . born in columbia hamilton county ohio morris attended the public schools . he studied law . he was admitted to the bar and commenced practice in batavia ohio . he served as clerk of the courts of clermont county . morris was elected as a democrat to the thirtieth congress to fill the vacancy caused by the death of thomas l .*

Class Mean Of Transp.: *german submarine u-32 was a type viia u-boat of nazi germany ' s kriegsmarine during world war ii . her keel was laid down on 15 march 1936 by ag weser of bremen as werk 913 . she was launched on 25 february 1937 and commissioned on 15 april with kapitänleutnant ( kptlt . ) werner lott in command . on 15 august 1937 lott was relieved by korvettenkapitän ( krv . kpt . ) paul büchel and on 12 february 1940 oberleutnant zur see ( oblt . z . s . ) hans jenisch took over he was in charge of the boat until her loss .*

Class Building: *the tootell house ( also called king ' s row or hedgerow ) is a house at 1747 mooresfield road in kingston rhode island that is listed on the national register of historic places . the two-story wood-shingled colonial revival house on a 3-acre ( 12000 m2 ) tract was designed by gunther and beamis associates of boston for mr . & mrs . f . delmont tootell and was built in 1932-1933 . house design was by john j . g . gunther and elizabeth clark gunther was the landscape architect for the grounds .*

Class Village: *angamoozhi is a village in pathanamthitta district located in kerala state india . angamoozhi is near seethathodu town . geographically angamoozhi is a high-range area . it is mainly a plantation township . both state run ksrtc and private operated buses connect angamoozhi to pathanamthitta city . tourist can avail the travelling facility by ksrtc service ( morning 5 30 from kumili and 11 30 from pathanamthitta ) in between kumili and pathanamthitta via vallakkadavu angamoozhi kakki dam and vadaserikkara and can enjoy the beauty of the forest . [ citation needed ]*

Table 11: DBPedia test examples correctly classified by EDUCE. Underlined set of words are excerpts extracted, one color per concept.

| Concept 0 | a top 50 company / the regular seasonal movement / the production of port wine / the american composers / a bollywood film / the third book / a secondary school / an american record executive personal manager promoter / a well-known v8 supercars presenter and commentator / a merchant and public official / a type viia u-boat / a house at / the main fort / a major poet |
|---|---|
| Concept 1 | born in philadelphia / tranquil star ( foaled / instructional documentary produced / bollywood film released in / nineteenth instalment in / secondary school for / born april 1 1958 / birth registered october–december 1896 / october 8 1804 - may 16 1875 / academy is the volleyball academy of / village in pathanamthitta |
| Concept 2 | ( 1800–1870 ) probably born in / ( 12 april / ) release in / is a bollywood film released in / an omnibus release from / a studio school located in / ( born april / gunji 8 july / ( october 8 1804 - may 16 / an iron collier in / of historic places . / fort ( also called |
| Concept 3 | top 50 company / medium-sized long-winged species with a bright rufous crown / endemic australian genus / album was released / bollywood film released / paul kemprecos and was published / inns of court exclusively entitled / debut album explores / lagarto ( lizard / hong kong cross-media creator / bird park in / wild blackwater river |
| Concept 4 | beltway around washington dc . / democratic republic of / plants in the / party live in / my texas . / asheville north carolina . / ohio . its / executive . he / democrat to the / south africa . / mooresfield road in kingston rhode island that / north to south . / kerala state india . |
| Concept 5 | is a top 50 company / is a genus of crane fly / is a red portuguese wine grape / is the debut studio record / is an abc family original movie / was a regional fashion entertainment and lifestyle publication / interactive technologies institute / is a beirut-based lebanese haute couture fashion designer / was a montenegrin serb boxer / is the chief technology officer / was a type viia u-boat / of historic places . the two-story wood-shingled / fort ( also / is a village |
| Concept 6 | company on the / bird of prey in / apple . in / documentary produced by / film released in / sequel to the / school for grades / animator . he / catcher . he / cross-media creator who / aircraft corporation to / museum was made in |
| Concept 7 | the brands club mahindra holidays club mahindra travel club / a bird of prey in / an endemic australian genus of around / snooker world rankings 1978/1979 the professional world rankings for / a black and white chinese animation short made in / a british rugby league periodical that / a russian youth football academy based in / a turkish cypriot singer and athlete . / a welsh rugby union and professional rugby league footballer of / a planter a confederate cavalry general in / a type viia u-boat of / a volleyball academy . / an uninhabited steep rocky island west of |
| Concept 8 | street sounds was / medium-sized long-winged species with a bright rufous / rataj places it / ep by linkin park / white chinese animation / british schools ( / secondary school for grades / argentine painter of the concretist and cubist schools / eminent anglican priest / hospital ships was built / national register of historic places / uninhabited steep rocky island west / village in pathanamthitta district |
| Concept 9 | translation pension fund for / adaptation which enables / mopeds produced by / instructional documentary produced by / bollywood film released in / book in the / river in kwazulu-natal / memoir of her / boxer in the / presidency of dmitry medvedev began / u-boat of nazi / harbour of helsingør dedicated to / uninhabited steep rocky island west of |

Table 12: Examples of excerpts that are extracted on DBPedia for all 10 concepts. Colors match the colors used in Table 11.

# D  VARYING THE NUMBER OF CONCEPTS ON DBPEDIA

Figures 4a and 4b (the second is a zoom to better see the differences between $10, 20$ and $50$ concepts) show the trade-off between concept accuracy and final accuracy on the DBPedia dataset, for multiple values of $C$ and different value of $\lambda$. Each line is a value of $C$. Each marker is a different value of $\lambda$, and values are (from left to right markers) $\{0, 0.01, 0.1, 0.5\}$. Separately, we show examplar excerpts of concepts learned with $C = 50$ and $\lambda = 0.5$ (purple dot in Figure 4b) in Table 13 (only 36 concepts are identified as present and we report those) to compare with the ones learned with $C = 10$ and $\lambda = 0.1$ in Table 11 (blue square in Figure 4b). With $C = 50$, the learned concepts qualitatively seem more consistent, even if the a posteriori concept accuracy is $81.37\%$ for the model with $C = 50$ which excerpts we report in Table 13, which is slightly less than the value ($83.90\%$) for the model with $C = 10$. Hence, learned concepts with $C = 50$ might be more similar to each others (e.g. less separable). Furthermore, an explanation with more concepts to look at and interpret is also harder to parse, which is why we consider that the binary intermediate representation should be low-dimensional. In future work, we would like to conduct a large scale experimentation involving human judgments to conclude on the interpretability benefits of different values of $C$. Note that with a too small value of $C$ (here, $C = 5$) final accuracy is harmed as the intermediate representation is too coarse.

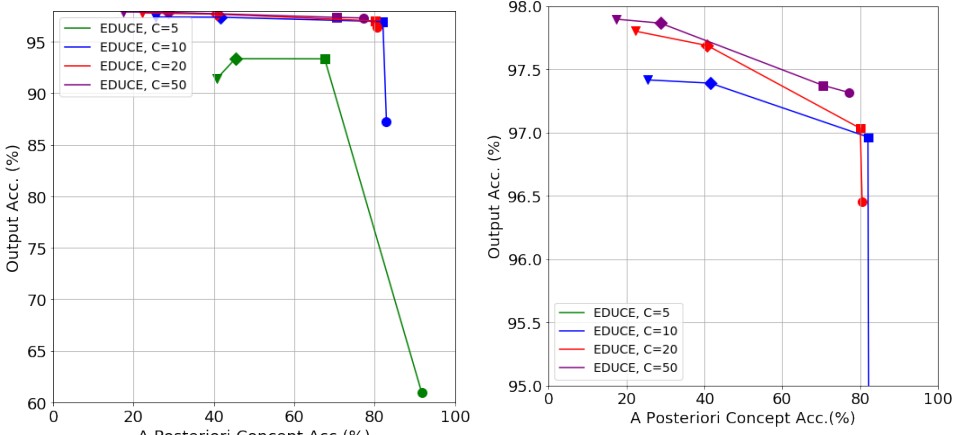

(a) DBPedia test performance, output accuracy vs a posteriori concept accuracy, for multiple values of $C$ and multiple values of $\lambda$.

(b) DBPedia test performance, output accuracy vs a posteriori concept accuracy, for multiple values of $C$ and multiple values of $\lambda$. Zoom between $95\%$ and $98\%$ final accuracy.

Figure 4: DBPedia test performance, output accuracy vs a posteriori concept accuracy, for multiple values of $C$ and multiple values of $\lambda$.

| Concept 0 | member of sire records act / primate native to / an indictment of policy changes / annexation of mrauk-u kingdom / regents of the university of / litterateur historian and activist / baseman first baseman and manager / democrat to the thirtieth congress to fill the vacancy caused / lieutenant commander henry crommelin / governor ' s residence and / prime minister of canada arthur meighen who held office / committee in ilam district |
|---|---|
| Concept 1 | is portuguese wine grape |
| Concept 2 | william james young ( 1800–1870 ) probably born in / april 1983 – 30 december 1992 ) was a british-bred / ramos ' crepidium ) is a member / brian wilson presents smile / film starring chad everett sarah / robert service then a professor / st . george ' s r . c . primary / vicky hamilton ( born april / james jim arthur bacon ( birth registered october–december / jonathan david morris ( october / clarence l . evans ( de-113 ) was a cannon-class / arthur goodson house |
| Concept 4 | a tribe of / a neighborhood in / a village in |
| Concept 6 | correspondents located in most of the district headquarters / forest on the western slopes on / river range in / album from stone temple / forest home of / seaside town of southwold and centers / whisky distillery . the distillery / favela of capão redondo in / bar and moved / hospital ships was built / house at 1747 mooresfield road / island west of the dingle peninsula . at / village in pathanamthitta district located |
| Concept 7 | shoes and other dance shoes and apparel . / racehorse who won at / his section macrophylli / players in the / man who flees the city and works in / biography of jesse lauriston livermore . the / painter who lived in / artist . hamilton is noted for managing the / footballer of the 1910s and ' 20s and coach of / singer from finland . elomaa was born in / exhibition at the / resident paid for / fan gyhirych in |
| Concept 9 | top 50 company / of the five operating divisions of the thomson corporation / rankings 1978/1979 the professional world rankings / collection of educational dvds tv series music / independent alternative newspaper / leading public university in the field of agriculture / network manager at the national / hong kong cross-media creator / electrical multiple units / artist-run centre located in vancouver bc with a vision / supply drinking water / centre of Ørskog municipality |
| Concept 10 | an international producer / last studio album / a 1971 road movie directed by monte hellman starring singer-songwriter / a damned poet / fryderyk chopin university of music / a british musician . he was a bassist / a french passport / a canadian record producer music publisher |
| Concept 11 | is a top 50 company on the / is portuguese wine grape that is used in / is a demo tape by / was first broadcast in syndication on / was a regional fashion entertainment and / is established under the / is an american record executive personal manager promoter and / was an american racecar driver . primarily / was a merchant and public official from / was a 4337-ton steamship built by / is a house at 1747 mooresfield road in / is located in / is a police station at |
| Concept 12 | an institute of the romanian academy / cultivar raised at the university / a guide to british schools / a secondary school / college oxford university he reviewed poetry / an american college / a degree in chemistry / a mars-orbiting aeronomy probe / is the volleyball academy / taluka of amravati |
| Concept 14 | university . the first of the / school and it was released on / school during the early 1990s specifically between / school for grades 9-12 located in / school later returning to / school . she graduated from dartmouth college in 1998 with / schools . he studied law . he was admitted to / university also known as evolutionsmuseet / high located in / skole which covers løken momoen |
| Concept 15 | bird of the arabian nights . / bird of prey in the falcon family . this bird / subfamily within the family euphorbiaceae . / hyena label in 2006 / faunistic studies and biogeography . / dolphins . he / mammal of africa / fossil collection in scandinavia / cat about to pounce . bruce |

| | |
|---|---|
| Concept 16 | first listed in the / considered by the / identified by melville / listed on the / delivered to nearly 8200 homes monday through / built in 1959 and has undergone some modifications from the / a 1965 us number 1 hit single for herman / powered by a miller four-cylinder . it / the former first lady of new / launched on 25 february 1937 and commissioned on / built in 1932-1933 . house design was by john / an uninhabited steep rocky island west of the |
| Concept 17 | provides public bus transportation |
| Concept 21 | 31 march 2014 pfzw / april 1983 – 30 december 1992 / april 27 1993 / july 6 1975 / 1933 to july 28 1934 / september 29 1962 / born april 1 1958 / birth registered october–december 1896 / born 5 august 1946 / november 18 1985 / december 29 1962 |
| Concept 22 | business since 1996 . / heliconiinae commonly called / name has been misapplied for / label in 1992 . / production created by / based in egremont / founded in 2005 / producer based in toronto who has / united states ) . / company thornaby-on-tees for burdick and cook london in / sweden established in / name could derive from |
| Concept 23 | newspaper comic strip reprints but later on original material / first described by / published his first description of it in / pornographic parodies of / was published in / newspaper reported in / romance novels from / published author himself as / newspaper reporter and editor and ( briefly ) a / motorcyclist magazine stated / a copy of / his novel on / first written notice about |
| Concept 24 | a church established / an historical text / a building of the trinity church / an american beaux-arts architect / 145th state house / a victorian era admiral-class battleship / a house at / a tourist attraction |
| Concept 27 | single-engined jet . the company / light golden color / list of motorcycles scooters and mopeds produced / hits released only / bollywood film released / whaling ship called / trains indian army personnel / american film and television screenwriter best / american racecar driver / united states navy commander and / type viia u-boat of nazi germany ' s kriegsmarine / submarine museum managed |
| Concept 28 | gecko as they claim their cars stick / bird of prey in the falcon family . this bird / subfamily within the family / japanese phantom thief / eagle . the avon eagles / brazilian butterfly swimmer / monoplane with wire braced wings / subsidiary of sparrow health / bird species including ducks great crested grebes swans and geese |
| Concept 29 | largest icelandic commercial banks / is the only mare / is the pacific ocean / water by the spoonful / agua fria high school / karnik ( devanagari / represents the creuse ' s 2nd constituency / a mars-orbiting aeronomy probe / is a large boulder / is an uninhabited steep rocky island west of the dingle / is situated on the lap of vindhya ranges |
| Concept 30 | a top 50 company / is portuguese wine grape / the original soundtrack on the sony bmg label / an american disney comedy franchise / a free independent alternative newspaper / the statutory and regulatory organization / an american record executive personal manager promoter / an american stock car racing driver / a grain buyer / high-speed electrical multiple units / an artist-run centre / to supply drinking water |
| Concept 31 | was only newspaper comic / is a selection / was released in 1986 . this particular release / are 1993 pornographic parodies / is the third book / was the original dj / is the eleventh / is a survey launch / is the world ' s tenth longest river |
| Concept 34 | british pottery manufacturing firm / a collection of american singer laura / italian film . it stars actor gabriele / damned poet and somewhat heartbreaking images / american comic strip creator / poker player pioneer poker theorist / female bodybuilding champion and pop singer / scientist astronomer and botanist |
| Concept 35 | ponderosa steakhouse and / is a medium-sized long-winged species with / is an endemic australian genus of around / is a venomous tree-dwelling snake . / is a brazilian butterfly swimmer . rooted in / is a family of / is a saltwater pond in |
| Concept 36 | a naive bayes / a bird of prey / an endemic australian genus / a venomous tree-dwelling snake / a young adult / an american sociobiologist / a brazilian butterfly swimmer / a large ruminant mammal / the largest and most important cave |

| | |
|---|---|
| Concept 37 | king street sounds / is an historical text / is a small rural public school district / was an american beaux-arts architect / george heard stone / s eighty-sixth house district / haven class of hospital ships / is a house / also called purandhar fort / is a village |
| Concept 38 | single-engined jet . the company was founded in 1998 by / mopeds produced by honda / chinese habitats that simulate generation ships / an argentine model vedette / sprint car driver he drove in the 1936 indianapolis / a confederate cavalry / u-boat of nazi germany ' s kriegsmarine during world / maritime museum kulturværftet and helsingør harbour / jeep underwater in the deepest |
| Concept 39 | species of tropical moss / endemic australian genus of around 20 species of flowering plants / lloyd mackenzie says "by blending the richness / swee ' pea / brookland plantation is plantation / plants and animals |
| Concept 40 | was the albuquerque new mexico-based manufacturer / is a member / is a collection of american singer / are 1993 pornographic parodies / is the third book / is an american record executive personal manager promoter / was a japanese master / was a u . s . representative / was a stalwart / is significant as a representative / is the most prominent feature |
| Concept 42 | is the second largest pension fund / is teen pop star aaron carter ' s first concert / is a bollywood film / is a 1923 roman à clef / is a secondary school / is an american actor film / is an american football guard / is the oklahoma senator / as a barracks ship / is a house / also called purandhar fort / is a separate campus |
| Concept 43 | film magic was founded on / rock monitor it is found in / meadows in the / video release overall released in / film released in 2010 which stars rishi / moon series written by / river in kwazulu-natal south / rock was the / film blazing saddles and for starring in the / lake county and / heights in the / family situated on the north slope of the / catchment within the murray–darling basin is located in the |
| Concept 46 | is an optical disc system developed by / is a selection made by / was released in 1986 . / is a video series by / is an omnibus release from / is the electronic music solo-project of / was on the / was a planned plug-in hybrid electric vehicle from / is a joint initiative by |
| Concept 47 | multi-jurisdictional lottery games / american thoroughbred race horse / rifle sport on ruthless / champion thoroughbred filly ruffian who went undefeated / british rugby league / a creek tennis courts track / french dancer choreographer / welsh rugby union and professional rugby league footballer / retired professional female bodybuilding champion / sport utility vehicle / galatasaray sports club / hole par-3 golf |
| Concept 48 | mikuláš ružomberok vrútky |
| Concept 49 | a company that published comic books beginning in 1933 . / an album by american comedian comedy writer and radio / a bollywood film released in 2010 which stars / the third book in the / an american comic strip creator who signed his work / a professional poker player pioneer poker theorist author of poker / a canadian lawyer professor politician and writer . / a british motorcycle made between 1959 and 1966 by phelon / a contemporary art gallery in melbourne australia . / a government-commissioned geographic study which forms the basis for the |

Table 13: Examples of excerpts that are extracted on DBPedia when using $C = 50$ concepts. Only 36 concepts are identified as present and we report those.

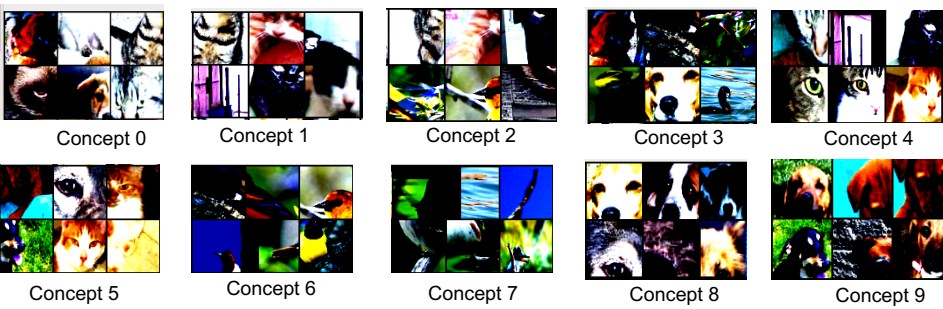

Figure 5: Samples of patterns extracted for each concept in the birds-cats-dogs dataset.

# E   APPLYING EDUCE PRINCIPLES FOR IMAGE CLASSIFICATION

Having assessed the relevance of our model on text data, we now turn to image data and explore if EDUCE is also able to extract meaningful concepts. In that case, the excerpts extracted in textual data are replaced by regions (or patches) in an image, and the recurrent neural architecture processing text is replaced by a classical convolutional neural network.

**Principles:**   Each image $x$ is first encoded through a convolutional neural network that output a representation in $h(x) = \mathbb{R}^{W \times H \times D}$ where $D$ is the number of output filters. Each vector $h_{i,j,.}(x) \in \mathbb{R}^D$ corresponds to a particular region $s$ of the input image (i.e the receptive field of the convolutional neural network). Then each $h_{i,j,.}$ is used to compute $p_\gamma(s|x,c)$ through a linear model and a softmax activation as in Section 2.1 (step 1), and to compute $p_\alpha(z_c = 1|s_c(x), c)$ (Section 2.1, step 2) here again with a linear model and a Bernoulli distribution. The semantics homogeneity is ensured by using an image classification network over the region captured by the receptive field in the previous convolutional neural networks that ensure that all regions captured from a particular concept "look the same" and look differently from the regions captured for other concepts.

**Experiments:**   To ensure the validity of our adaptation of EDUCE to images, we conduct very simple experiments on a dataset composed of $224 \times 224$ RGB images split in 3 categories: dogs, cats, and birds[6] in equal proportion. We train on $3,000$ images and test on $3,000$ images. We build our model on top of a pretrained VGG-11 model Simonyan & Zisserman (2014) which is used for computing the intermediate representations $h(x)$. Said otherwise, the output of the VGG-11 model is used for both detecting relevant regions and thus building the binary representation $z$, but also to ensure the homogeneity of extracted regions during training. Figure 5 shows extracted patterns and associated categories. Final classification performance is $91.6\%$ with 10 concepts. In Figure 5 we plot extracted patterns for the 10 concepts and report in Figure 6a the weights of the final classifier. From these two figures, we can interpret the model's behavior: concept 8 and concept 9 show what differentiates a *dog* from a *cat* or a *bird*, and support the classifier's prediction of the dog category. Moreover, by looking at which regions are extracted by our model (Figure 6b ), we can see that EDUCE can focus its attention to discriminative regions of the images.

---

[6]We construct this dataset by combining random images from the Caltech Bird 200-2011 dataset (Wah et al., 2011) with images of the cats-and-dogs Kaggle dataset (Kaggle, 2013)

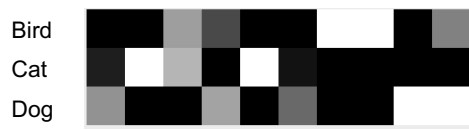

(a) Final Classifier weights, each column is a concept (0 to 9, from left to right) each row a final category. The lighter, the higher the weight is. Concepts 6 and 7 are associated to category 'Bird', Concept 1 and 4 to 'Cat' and concepts 8 and 9 to 'Dog'.

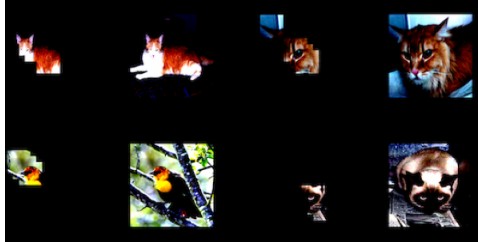

(b) Examples of classified images (column 2 and 4) and corresponding extracted patterns (column 1 and 3) for concept detected as present.

Figure 6: Interpretation of EDUCE on images.

