# OpenReview forum: "EDUCE: Explaining model Decision through Unsupervised Concepts Extraction"
_ICLR.cc/2020/Conference — Reject_

### Official Review · AnonReviewer3 · 2019-10-22
**Official Blind Review #3**

**Rating:** 6

**Review:**

The paper introduces a new concept-based interpretability method that lies in the family of self-interpretable models (i.e. it's not a post-hoc method). Self-interpretability is achieved by a two-stage model: First, a concept-extractor finds the related pieces of consecutive words (excerpts) in a given text that are related to a concept among a set of given concepts (if any), then the model makes its predictions solely based on the presence or absence of concepts (binary).  The most useful part of the algorithm is that there is no need for concept annotations. The work then experimentally shows that their method, although does not outperform a non-self-interpretabile baseline but has better performance and interpretation compared to rival methods.
The paper is quite well-written. The introduced method is well-justified and the use of concept-based self-interpretable models is very useful to the field. It is also interesting to see that the performance is comparable to non-self-nterpretable baselines which would make a case for the use of self-interpretable models. I have two main concerns with this work. First, there is the novelty concern. The idea of unsupervised extraction of concepts for interpretability was introduced before (https://arxiv.org/pdf/1902.03129.pdf) and is not discussed by the authors (although the utilized terminology is very similar).  Authors should make a much more comprehensive discussion of what already exists in the concept-based interpretability literature and make the contribution of this work more clear (unsupervised concept extraction for a self-interpretable model instead of post-hoc interpretations). Secondly, although some of the objective experimental results (BeerAdvocate results, to some degree) suggest that the method is indeed capable of discovering meaningful, consistent, and separable concepts on its own, there is not enough discussion of why it actually should be the case. It's very easy to assume the introduced training procedure to extract separable but meaningless concepts(i.e. the excerpts of a concept are separable from that of other concepts and are consistent with each other in the eyes of the network while they are not consistent with a concept in the eye of human rationale; both loss terms will be minimized but there is no human-interpretability. One big issue with the experiments section is the A Posteriori Concept Acc metric as it only measures the separability and consistency of discovered concepts; first of all, it can be high while the extracted concepts are meaningless to humans, secondly, using it as a measure of comparison to rival methods is not quite fair as the introduced method is directly optimizing for this metric. The subjective results are interesting but not convincing. Adding a section for human subject experiments (following the previous work in concept-based interpretability) would make the results section much crisper. It also seems unlikely that all of the discovered concepts follow the desiderata and the fact that this is not mentioned makes it even more difficult to assess the amount of cherry-picking in the mentioned subjective results; there should be a discussion of cases of failure and why it happens. My score will change accordingly as the authors address the raised issues.

A few questions and suggestions:

* It would be more interesting to show subjective results of the model for cases of mistake; does the interpretability make sense when the model's prediction is wrong?

* It seems necessary to discuss how much the results are sensitive to selecting C? If the performance is robust, it would be interesting to see what happens for a large C? Does it discover more fine-grained concepts or most of the additional concepts would be null?

* For the model architecture, why did the authors choose to use the straight-through estimator and not other methods (e.g. concrete layers, ...)

* Some of the figures should be larger (much larger)

Thanks again for a well-organized and easy-to-follow paper.

**Experience Assessment:**

I have published in this field for several years.

**Review Assessment: Checking Correctness Of Derivations And Theory:**

N/A

**Review Assessment: Checking Correctness Of Experiments:**

I carefully checked the experiments.

**Review Assessment: Thoroughness In Paper Reading:**

I read the paper thoroughly.

---

> ### Author Response · Authors · 2019-11-08
> **Reply to Reviewer 3**
>
> Thank you very much for your feedback, we are happy that you found the paper clear and easy-to-follow. Your comments are very relevant and helped us clarify the paper. We have modified the paper thanks to your points, and below you will find detailed answers to your concerns.
>
> Regarding the related work on concept discovery https://arxiv.org/pdf/1902.03129.pdf:
>
> You are very right that this is a relevant work and that the terminology is similar. We should have included it in our literature and we have modified the paper to include it and emphasize the differences with our work (see Section 3 on related work). First, the model in https://arxiv.org/pdf/1902.03129.pdf is trained a posteriori, to help understanding what is learned by a neural model, while EDUCE is trained end-to-end to classify solely based on the discovered concepts. This is a major difference since our approach aims at discovering concept which presence/absence is discriminant resulting in a fully interpretable model. Second, we focus on text data (even if an extension to image is provided in our supplementary material) while they interpret a model trained on images.
>
> Concerning the A posteriori Concept Accuracy measure:
>
> We agree that formally, concept accuracy only measures (i) concept consistency (ii) concept separability with respect to a given concept classifier architecture, and is not guaranteed to correlate with human judgments. We now made that point clearer in the paper in the paragraph “Metric” of Section 4. Indeed, the correlation between the extracted concepts and human defined concepts mainly depends on the inductive bias captured by the underlying loss function. Evaluating how well discovered concept align with human defined concept is a general challenge in learning explainable/interpretable models in an unsupervised setting. Typically, for images, one assumption is to consider that convex regions of pixels carry human interpretable information (as in the LIME model). In our case, we believe it is a reasonable assumption to employ a linear concept classifier over pre-trained word embeddings such that it linear separates the word embedding space, and words embeddings have been shown to carry words’ semantics (see for example the improvements in downstream semantics tasks in Mikolov et al. https://arxiv.org/pdf/1301.3781.pdf). This motivates both the EDUCE model and the use of the A posteriori Concept Accuracy metric. This aspect is now better discussed at the end of Section 2 where we present the model, and Section 4 where we present the metric. We also included in supplementary material Table 10 the concepts extracted by the No Concept Loss model, to show that they form less semantically consistent units and are hard to interpret, and mention this in Section 4 paragraph “Interpreting EDUCE on AGNews”.
>
> Indeed, ideally if human labels are available we could confront our concepts with human judgments. This is where EDUCE presents advantages: EDUCE employs only a few concepts, that can easily be described by the set of excerpts extracted for that concept. Therefore, we could design an experiment where human subjects judge of the semantics of EDUCE’s discovered concepts by looking at the set of extracted excerpts for each concept. Such a study is time and money-consuming and is currently considered as a direction for future work. We experiment on the Beer that includes human rationales as proxies to analyze if our model is able to focus on some relevant part of input texts. This is not fully satisfactory since it is not directly evaluating the relevance of concepts, but it gives insights on the ability of EDUCE to capture meaningful concepts, and relevant associated excerpts.
> At last, please also consider that our model could easily benefit from a partial supervision (i.e human rationales with labels) to semi-supervise the concept extraction process, which we consider as a future study.

---

> > ### Author Response · Authors · 2019-11-08
> > **Reply to Reviewer 3 (continued)**
> >
> > We replied to part of your concerns in the previous comment. We continue here.
> >
> > Regarding concept’s discovery failure cases:
> >
> > Please note that all reported results in the paper are computed with the best seed and hyper-parameters cross-validated on final and concept accuracies, and not interpretability — we did not perform cherry-picking and used classical model selection techniques.
> >
> > As we show in the paper, discovered concepts seem to align with concepts a human would define in the case of the AGNews, SST and Beer dataset. We also report the concepts discovered and some interpretation examples for the DBPedia dataset in Table 11 and 12 in the Supplementary material showing that if some concepts are easy to map to human notions (e.g. concept 1 relates to birth/birthdate and numbers, concept 4 captures locations) some concepts are really hard to parse (e.g. concept 0 is consistent only in the fact that all extracted patterns start with “a” or “the” and concept 7 is similar, and the two concepts are separable because they are detected as present in different classes). One way to avoid this behaviour would be to benefit of excerpts label provided by humans at train time, what can be easily handled in our loss function and which is a nice property of our method.
> >
> > Regarding final classification failure cases:
> >
> > We added failure examples in the Supplementary material. Table 8 shows these failure cases of the final classifier. In some of these, the predicted class could be considered as an appropriate label. Importantly, we see that in these cases the concepts that are identified as present in the text are relevant to the set of concepts that the model automatically defined. We added that point in the paper section 4 paragraph “Interpreting EDUCE on AGNews”.
> >
> > Below we reply to each of your other comments and questions:
> >
> > * Effect of the number of concepts C:  Figure 2 (which we made larger) shows output accuracy versus a posteriori concept accuracy on the SST dataset, for different number of concepts C and different values of λ. We added in the paper the following analysis that was missing in Section 4, when we discuss the results on SST: For a fixed number of concepts C, choosing the adequate value of λ allows the user to balance between final accuracy and concepts’ accuracy. Therefore similar performances can be achieved by different number of concepts C, each with an adequate value λ: C=20 with λ = 0.1 (dotted purple line, squared marker) achieves ~41.5% final accuracy and ~70% concept accuracy, while C=5 with λ = 0.01 (plain blue line, diamond marker) achieves ~42.0% final accuracy and ~60% concept accuracy. The smaller the number of concepts, the smaller the value of λ which is expected as with less concepts the task of the concept classifier is easier. Note however than a too-small number of concept can harm final accuracy if the binary representation is too coarse by having to few dimensions. We launched additional experiments with multiple number of concepts on the DBPedia dataset. It will take days to be finished and be analyzed, but we will include it in the camera ready version should the paper be accepted.
> >
> > *Model architecture: We chose to use the straight-through estimator for its simplicity, but relaxation such as Gumbel-softmax could be used.
> >
> > * Larger figures: we have enlarged figures in the paper.
> >
> > Thank you again for a very valuable feedback, please let us know if you have any other suggestions to include in the paper.

---

> > > ### Comment · AnonReviewer3 · 2019-11-13
> > > **Rebuttal**
> > >
> > > Thanks for your detailed response. I really enjoyed the discussion of failure cases. I have changed my score accordingly. I cannot lean towards full acceptance of the work as the ultimate verification of the method (human-subject tests) are not provided. The paper is still missing a comprehensive discussion of and a detailed comparison with existing concept-based interpretability methods. Right now it's briefly discussed in the "A posteriori explanations." paragraph which is first, a very general title. Apart from Ghorbani et al and Kim et al's work, Zhou et al. and Bau et al.'s works should also be mentioned and discussed. The field is small and young and everyone could benefit.

---

> > > > ### Author Response · Authors · 2019-11-14
> > > > **Rewriting of the related work section + included mentioned works**
> > > >
> > > > Thank you reply, we are happy that the failure cases discussion met your expectations.
> > > >
> > > > Thank you for pointing out the related works. We have rewritten the related work section, and separated concept-based (either a posteriori of self-interpretable) models from the rest of the literature review. We included Zhou et al. and Bau et al. works.
> > > >
> > > > Note that Zhou et al. [2] and Bau et al. [1] require human labels at the concept level (they experiment on the Broden [1] dataset) while EDUCE, as Ghorbani et al., is unsupervised. Moreover, Zhou et al. and Bau et al. methods explain an already trained model (which we refer to as a posteriori explanations) while EDUCE is learned end-to-end to be interpretable. Finally, we focus on text data and the aforementioned works experiment on image data.
> > > >
> > > > It would be very interesting to conduct human experiments to evaluate EDUCE's concepts. Such evaluation takes time and is costly. In our paper, we experiment on the Beer dataset as a proxy for such evaluation (see our detailed reply on this point to Reviewer 2) but in our future works, we will try to include such evaluation.
> > > >
> > > > [1] David Bau, Bolei Zhou, Aditya Khosla, Aude Oliva, and Antonio Torralba. Network dissection: Quantifying interpretability of deep visual representations. In CVPR 2017.
> > > >
> > > > [2] Bolei Zhou, Yiyou Sun, David Bau, and Antonio Torralba. Interpretable basis decomposition for visual explanation. In ECCV, 2018.

---

### Official Review · AnonReviewer2 · 2019-10-23
**Official Blind Review #2**

**Rating:** 3

**Review:**

This paper presents a classification model focused on interpretability. The model, Explaining model Decision through Unsupervised Concepts Extraction (EDUCE), is applied to a text classification task, while the authors argue in the appendix that this is also applicable to a wider problem, such as image classification.

The model is composed of three parts: the first part is detecting salient spans of text relevant to the text classification problem, the second part assigning each salient span a concept label, and the third part which does the classification task based on the binary concept feature label. The models’ loss is composed of two parts: (A) minimizing the cross-entropy of text classification loss and (B) minimizing the cross-entropy of concept classification loss. For (A), as the first and second part of the model introduce discrete choices, they use a RL with Monte-Carlo approximation of gradients.

The system is evaluated under two measure: 1) classification accuracy and 2) concept accuracy. They define the concept accuracy as follows: after training, they train a classifier that takes output (in the form of <salient span, their concept label>) of the model from the test portion of the data. They split this output into train and test, and report the test accuracy. This aims to show how consistent is the labeling of the salient spans for different methods: if the concept label set correctly merged together semantically similar spans, this “concept accuracy” would be higher. This is a new metric they are proposing. While it is interesting, I would like to see *some* studies on how this correlates with human’s judgements on how interpretable the model is. The paper is introducing a new measure *and* new model, and it’s hard to be persuaded the model is doing well based on this new measure, when there is little ground to know what this measure really measures.

Overall, I’m not impressed with the models’ performances. The aspect rationale annotated beer sentiment dataset, presented by Lei et al (2016), has provided one of few opportunities to evaluate interpretability / rationale model quantitatively. The paper evaluates on this measure, which is included in the appendix, and the results are pretty disappointing compared to the existing models such as Lei et al’s initial baseline or Bastings et al. While the paper argues this method isn’t necessarily designed for this task unlike the other methods, I’m not sure this is necessarily the case. Bastings et al could be applied to other tasks that model is evaluated on, such as DBPedia and AGNews classification. The difference comes on how easy it is to interpret the methods, as these other rationale-based text processing methods would make use of captured words, while EDUCE would make use of detected “concept” clusters. Currently, the only real baselines are the ablations of its own model.

Table 3 is quite interesting, different “concepts” capture different aspects fairly well.

Not having a concept loss actually helps the classification accuracy. Would the concepts learned without concept loss qualitatively very different? This goes back to my original point that their new measure of "concept accuracy" is vague.

Other comments and Q:
- Figure (3), the visualization is a bit confusing cause it is unclear whether it is each span is a set of spans or a single span. Also, I would recommend making figures colorblind friendly, if possible.
Q: what kind of classifier was used for the evaluation metric “concept accuracy” classifier? I don’t think it’s mentioned.
Q: why are you sampling a test set for DBPedia experiments? Is it for efficiency reason?
Q: how sensitive is model’s performance to the hyper parameters, especially the number of concepts?
Q: the current baseline classifier is a simple BiLSTM one, which definitely perform a lot worse than recent pre-trained LMs such as BERT. Would it be easy to use this method on top of richer representation such as pertained LM outputs?
Q: how would this connects to saliency map literature in computer vision? I guess these would be mostly “a posteriori” explanations? Discussion would be helpful.


**Experience Assessment:**

I have read many papers in this area.

**Review Assessment: Checking Correctness Of Derivations And Theory:**

I assessed the sensibility of the derivations and theory.

**Review Assessment: Checking Correctness Of Experiments:**

I assessed the sensibility of the experiments.

**Review Assessment: Thoroughness In Paper Reading:**

I read the paper thoroughly.

---

> ### Author Response · Authors · 2019-11-08
> **Reply to Reviewer 2**
>
> Thank you for your valuable feedback, we found your comments to be relevant and helpful in improving the paper. We have modified the paper accordingly, and below you will find detailed replies to your concerns.
>
> Concerning the A posteriori Concept Accuracy measure:
>
> We agree that formally, concept accuracy only measures (i) concept consistency (ii) concept separability with respect to a given concept classifier architecture, and is not guaranteed to correlate with human judgments. We now made that point clearer in the paper in the paragraph “Metric” of Section 4. Indeed, the correlation between the extracted concepts and human defined concepts mainly depends on the inductive bias captured by the underlying loss function. Evaluating how well discovered concept align with human defined concept is a general challenge in learning explainable/interpretable models in an unsupervised setting. Typically, for images, one assumption is to consider that convex regions of pixels carry human interpretable information (as in the LIME model). In our case, we believe it is a reasonable assumption to employ a linear concept classifier over pre-trained word embeddings such that it linear separates the word embedding space, and words embeddings have been shown to carry words’ semantics (see for example the improvements in downstream semantics tasks in Mikolov et al. https://arxiv.org/pdf/1301.3781.pdf). This motivates both the EDUCE model and the use of the A posteriori Concept Accuracy metric. This aspect is now better discussed at the end of Section 2 where we present the model, and Section 4 where we present the metric. We also included in supplementary material Table 10 the concepts extracted by the No Concept Loss model, to show that they form less semantically consistent units and are hard to interpret, and mention this in Section 4 paragraph “Interpreting EDUCE on AGNews”.
>
> Indeed, ideally if human labels are available we could confront our concepts with human judgments. This is where EDUCE presents advantages: EDUCE employs only a few concepts, that can easily be described by the set of excerpts extracted for that concept. Therefore, we could design an experiment where human subjects judge of the semantics of EDUCE’s discovered concepts by looking at the set of extracted excerpts for each concept. Such a study is time and money-consuming and is currently considered as a direction for future work. We experiment on the Beer that includes human rationales as proxies to analyze if our model is able to focus on some relevant part of input texts. This is not fully satisfactory as detailed in the next section, but it gives insights on the ability of EDUCE to capture meaningful concepts, and relevant associated excerpts.
> At last, please also consider that our model could easily benefit from a partial supervision (i.e human rationales with labels) to semi-supervise the concept extraction process, which we consider as a future study.
>
> Concerning our results on the Beer Dataset experiment:
>
> The rationales are extracted a priori without the notion of concepts. As we mention in the Supplementary Section B.2 where we analyse the results of the experiment on predicting each aspect’s score separately, manual inspection shows us that in the case of the Palate aspect (where we report poor performance), the model defines a concept dedicated to the notion of numbers (e.g. corresponding to excerpts “12 ounce bottle”) that is consistent, yet not part of the gold rationale. We believe there have been a misunderstanding due to a poor wording in the paper: by saying rationales model are “are specifically designed for this task”, we meant that they are designed to extract rationales for the prediction. We modified this sentence to make it clearer. As you correctly point out, EDUCE uses automatically defined concepts, based on excerpts, and is not concerned with rationale extraction per se. In the case of the Beer Dataset, we believe the notion of concepts best matches the different aspects, in the case of giving an overall score. This is why we emphasize on the regression experiment of predicting the 4 aspects and the overall score at once, and Table 2 and 3 show that in this experiment some concepts are capturing a specific aspect. We cannot compare in this experiment to Lei et al. (2016); Bastings et al. (2019) as to the best of our knowledge, they do not report their rationales retrieval performance when predicting the 4 aspects and the overall score at once (but only the MSE).
>
> In the next comment, we reply to each of your other comments and questions.

---

> > ### Author Response · Authors · 2019-11-08
> > **Reply to Reviewer 2 (continued)**
> >
> > In the previous comment we replied to your main concerns, below we reply to each of your other comments and questions:
> > * Figure 3: each span of each color consists of a single span (each concept can appear at most once in the input text) and one color matches the span for one concept.
> >
> > * A posteriori concept classifier: For the concept accuracy metric we use the same architecture as EDUCE’s concept classifier (linear classifier without bias). We changed the paper to mention it clearly.
> >
> > * DBPedia dataset: yes we do subsample the dataset for efficiency reasons, as DBPedia was specifically introduced as “large-scale dataset” in Zhang et al., 2015.
> >
> > * Effect of the number of concepts C:  Figure 2 (which we made larger) shows output accuracy versus a posteriori concept accuracy on the SST dataset, for different number of concepts C and different values of λ. We added in the paper the following analysis that was missing in Section 4, when we discuss the results on SST: For a fixed number of concepts C, choosing the adequate value of λ allows the user to balance between final accuracy and concepts’ accuracy. Therefore similar performances can be achieved by different number of concepts C, each with an adequate value λ: C=20 with λ = 0.1 (dotted purple line, squared marker) achieves ~41.5% final accuracy and ~70% concept accuracy, while C=5 with λ = 0.01 (plain blue line, diamond marker) achieves ~42.0% final accuracy and ~60% concept accuracy. The smaller the number of concepts, the smaller the value of λ which is expected as with less concepts the task of the concept classifier is easier. Note however than a too-small number of concept can harm final accuracy if the binary representation is too coarse by having to few dimensions. We launched additional experiments with multiple number of concepts on the DBPedia dataset. It will take days to be finished and analysed, but we will include it in the camera ready version should the paper be accepted.
> >
> > * BERT: Our model could be used on top of pre-trained model such as BERT. We used biLSTM as our goal was to compare with the non-interpretable counterpart rather than achieve state-of-the art classification results, but also to use a similar architecture as in Lei et al. (2016); Bastings et al. (2019).
> >
> > * Saliency maps: In the supplementary material, we propose an extension of EDUCE to image classification, where concepts corresponds to patches extracted from the images. It is linked to the saliency map literature, with two differences: Here our saliency maps are composed of patches associated to concepts, and the different patches are encouraged to be homogenous per concept thanks to the concept loss.

---

### Official Review · AnonReviewer1 · 2019-10-23
**Official Blind Review #1**

**Rating:** 8

**Review:**

The authors proposed a self-explainable deep net architecture that could be used for text categorization. The main idea is to force the network to extract "excerpts", from the input text, each corresponds to a concept, which are also learned for interpretation. The classification is finally made based off of the learned concept, which is a binary vector. All three steps are learned in an end-to-end manner. The learning of concepts is regularized to make sure the concepts are consistent and non-overlapping. The idea sounds interesting and the experimental results support the usefulness of the proposed method on a variety of datasets. My sole concern is about the sensitivity analysis of the explanation, i.e. how robust is the explanation with respect to the perturbations that do not change the classifier prediction. It has been discussed in the literature that many explanation methods suffer from this sensitivity issue.

**Experience Assessment:**

I have read many papers in this area.

**Review Assessment: Checking Correctness Of Derivations And Theory:**

I assessed the sensibility of the derivations and theory.

**Review Assessment: Checking Correctness Of Experiments:**

I assessed the sensibility of the experiments.

**Review Assessment: Thoroughness In Paper Reading:**

I read the paper at least twice and used my best judgement in assessing the paper.

---

> ### Author Response · Authors · 2019-11-08
> **Reply to reviewer 1**
>
> Thank you for your positive feedback on our paper. We agree that it would be very interesting to conduct a sensitivity analysis of the model (see responses to other reviewers). Such analysis would require more time and we will consider conducting one in our future work.

---

### Author Response · Authors · 2019-11-08
**Paper update and replies**

We thank the reviewers for their valuable feedback. We have replied to each reviewer's concerns separately, and have modified the paper accordingly. Reviewers 2 and 3 had similar questions about the a posteriori accuracy measure and the effect of the number of concepts, so these parts of each reply are similar.

---

> ### Author Response · Authors · 2019-11-15
> **Paper update with multiple values of C on DBPedia**
>
> As mentioned to Reviewer 2 and 3, we have conducted an analysis of the effect of varying the number of concepts C on DBPedia (in the original submission we were doing so for SST dataset). The results are reported in Supplementary material section D. To summarize, when comparing C=10 to C=50 concepts on DBPedia, with C=50 the learned concepts qualitatively seem more consistent, even if the a posteriori concept accuracy is 81.37% for the model
> with C = 50 which excerpts we report in Table 13, which is slightly less than the value (83.90%) for
> the model with C = 10. Hence, learned concepts with C = 50 might be more similar to each others (i.e. less separable). Furthermore, an explanation with more concepts to look at and interpret is also harder to parse, which is why we consider that the binary intermediate representation should be low-dimensional. Note that with a too small value of C (e.g. C = 5) final accuracy is harmed as the intermediate representation is too coarse.

---

### Decision · Program_Chairs · 2019-12-19

**Decision:**

Reject

**Comment:**

This paper introduces a method for building interpretable classifiers, along with a measure of "concept accuracy" to evaluate interpretability, and primarily applies this method to text models, but includes a proof of concept on images in the appendix.

The main contributions are sensible enough, but the main problems the reviewers had were:
A) The performance of the proposed method
B) The lack of human evaluation of interpretability, and
C) Lack of background and connections to other work.

The authors improved the paper considerably during the rebuttal period, and might have addressed point C) satisfactorily, but only after several back and forths, and at this point it's too late to re-evaluate the paper.  I expect that a more polished version of this paper would be acceptable in a future conference.

I mostly ignored R1's review as they didn't seem to put much thought into their review and didn't respond to requests for clarifications.